# The photosystem I supercomplex from a primordial green alga *Ostreococcus tauri* harbors three light-harvesting complex trimers

**Asako Ishii**[1†], **Jianyu Shan**[2,3†], **Xin Sheng**[2‡], **Eunchul Kim**[1,4], **Akimasa Watanabe**[1,4], **Makio Yokono**[1,4], **Chiyo Noda**[1], **Chihong Song**[5,6], **Kazuyoshi Murata**[5,6], **Zhenfeng Liu**[2,3]*, **Jun Minagawa**[1,4]*

[1]Division of Environmental Photobiology, National Institute for Basic Biology, Okazaki, Japan; [2]National Laboratory of Biomacromolecules, CAS Center for Excellence in Biomacromolecules, Institute of Biophysics, Chinese Academy of Sciences, Beijing, China; [3]College of Life Sciences, University of Chinese Academy of Sciences, Beijing, China; [4]Department of Basic Biology, School of Life Science, the Graduate University for Advanced Studies, Okazaki, Japan; [5]National Institute for Physiological Sciences, National Institutes of Natural Sciences, Okazaki, Japan; [6]Exploratory Research Center on Life and Living Systems (ExCELLS), National Institutes of Natural Sciences, Okazaki, Japan

**\*For correspondence:**
liuzf@ibp.ac.cn (ZL);
minagawa@nibb.ac.jp (JM)

[†]These authors contributed equally to this work

**Present address:** [‡]Shenzhen Jingtai Technology Co. Ltd, Shenzhen, China

**Abstract** As a ubiquitous picophytoplankton in the ocean and an early-branching green alga, *Ostreococcus tauri* is a model prasinophyte species for studying the functional evolution of the light-harvesting systems in photosynthesis. Here, we report the structure and function of the *O. tauri* photosystem I (PSI) supercomplex in low light conditions, where it expands its photon-absorbing capacity by assembling with the light-harvesting complexes I (LHCI) and a prasinophyte-specific light-harvesting complex (Lhcp). The architecture of the supercomplex exhibits hybrid features of the plant-type and the green algal-type PSI supercomplexes, consisting of a PSI core, an Lhca1-Lhca4-Lhca2-Lhca3 belt attached on one side and an Lhca5-Lhca6 heterodimer associated on the other side between PsaG and PsaH. Interestingly, nine Lhcp subunits, including one Lhcp1 monomer with a phosphorylated amino-terminal threonine and eight Lhcp2 monomers, oligomerize into three trimers and associate with PSI on the third side between Lhca6 and PsaK. The Lhcp1 phosphorylation and the light-harvesting capacity of PSI were subjected to reversible photoacclimation, suggesting that the formation of *Ot*PSI-LHCI-Lhcp supercomplex is likely due to a phosphorylation-dependent mechanism induced by changes in light intensity. Notably, this supercomplex did not exhibit far-red peaks in the 77 K fluorescence spectra, which is possibly due to the weak coupling of the chlorophyll *a*603-*a*609 pair in *Ot*Lhca1-4.

## Editor's evaluation

This fundamental work represents an important contribution to our understanding of the diversity of photosynthetic mechanisms across the branches of phototrophic life, with the first high-resolution structure (2.9 Å) of a photosynthetic complex from the green alga, *Ostreococcus tauri*, an ecologically important green alga utilizes a unique antenna complex, Lhcp. The evidence suggests mechanism found here is distinct from the classical antenna state transitions seen in other organisms studied thus far, expanding our knowledge of how photosynthetic systems react to changes in light

conditions and leading to a new understanding of the function and evolution of light-harvesting antennas.

## Introduction

Phytoplankton are the major primary producers in the aquatic environments, and provide organic matter for marine food webs by converting solar energy into chemical energy through photosynthesis. As a member of natural phytoplankton living in the ocean, *Ostreococcus tauri* is a unicellular green alga widespread in marine environments and crucial for the aquatic ecosystem (*Derelle et al., 2006*; *Palenik et al., 2007*). It is also known as the smallest free-living eukaryote (*Courties et al., 1994*) and belongs to Prasinophyceae, a class of green algae at the near basal position in the evolution of green lineage and most closely related to the first green alga known as 'ancestral green flagellate' (*Lewis and McCourt, 2004*).

A prasinophyte-specific Lhc protein named Lhcp was found in *O. tauri* (*Six et al., 2005*; *Swingley et al., 2010*) and *Mantoniella squamata* (another member of Prasinophyceae) (*Jiao and Fawley, 1994*; *Schmitt et al., 1994*). There are two classes of Lhcp (Lhcp1 and Lhcp2) in addition to the common LHC proteins, namely six Lhca proteins (Lhca1-6) and two minor monomeric Lhcb proteins (Lhcb4 and Lhcb5), serving as the peripheral antennae of *O. tauri* PSI and PSII (*Six et al., 2005*; *Swingley et al., 2010*). The Lhcp proteins form a highly abundant antenna complexes in *O. tauri* (*Swingley et al., 2010*), and the carotenoid composition of the OtLhcp complexes is largely different from those of LHCIs or light-harvesting complexes II (LHCIIs) in plants and other green algae (such as a core chloro-phyte *Chlamydomonas reinhardtii*) (*Minagawa, 2009*). While Lhcp, Lhcb (apoproteins of LHCII), and Lhca (apoproteins of LHCI) all belong to the LHC superfamily, Lhcp proteins form a separate clade with characteristics of an ancestral state of LHC proteins instead of belonging to the clades of Lhcb or Lhca (*Six et al., 2005*). Moreover, the organization of pigment molecules within the Lhcp complexes exhibits distinct features in comparison with plant LHCII according to a previous spectroscopic study (*Goss et al., 2000*).

Under low light (LL) conditions, the Lhcp complexes of *O. tauri* can assemble with the PSI-LHCI to form a larger PSI-LHCI-Lhcp supercomplex, whereas the amount of PSI-LHCI-Lhcp supercomplex is greatly reduced under high-light (HL) conditions (*Swingley et al., 2010*). It remains unclear how the Lhcp complexes assemble with PSI and establish energy transfer pathways with the interfacial PSI subunits under LL conditions. The arrangement of various Chl and carotenoid molecules within the Lhcp complexes is also unknown and awaits to be analyzed through further studies.

## Results and discussion
### The PSI-LHCI-Lhcp supercomplex

To stabilize the photosynthetic supercomplexes from *O. tauri* during purification process, the detergent-solubilized thylakoid membrane was treated with amphipol A8-35, which is an amphipathic polymer to substitute for detergents, before the sucrose density gradient (SDG) ultracentrifugation (*Figure 1a*, *Figure 1—figure supplement 1a*), according to the previous protocol successfully employed to stabilize the PSII supercomplex from *C. reinhardtii* (*Watanabe et al., 2019*). Pigment composition analysis of the A3L fraction demonstrates that the most abundant light-harvesting carotenoid is prasinoxan-thin (Prx, which is unique to prasinophytes), among other carotenoids of lesser abundance, including dihydrolutein (Dlt) and micromonal (*Figure 1—figure supplement 1c*, *Table 1*). Although the typical Chls (such as Chl *a* and Chl *b*) and carotenoids (violaxanthin/Vio, 9'-cis-neoxanthin/Nex and β-caro-tene) found in green plants were also present in *O. tauri*, lutein (a major carotenoid found in plant LHCII) was not detected. Polypeptides for the PSI core and its peripheral antennae in the A3 and A3L fractions were characterized by SDS-PAGE (*Figure 1—figure supplement 1b*) and mass spectrometry analysis (*Supplementary file 1a*). These findings indicate that A3 is primarily composed of polypep-tides for PSI-LHCI supercomplex, whereas A3L includes Lhcp1 and Lhcp2 alongside the constituents present in A3, along with comigrated PSII polypeptides, which is essentially in agreement with a previous report (*Swingley et al., 2010*).

Fluorescence properties of A3 and A3L were characterized by fluorescence decay associated spectra at 77 K (FDAS, *Figure 1—figure supplement 2a–c*). The fluorescence lifetimes of A3 were mainly

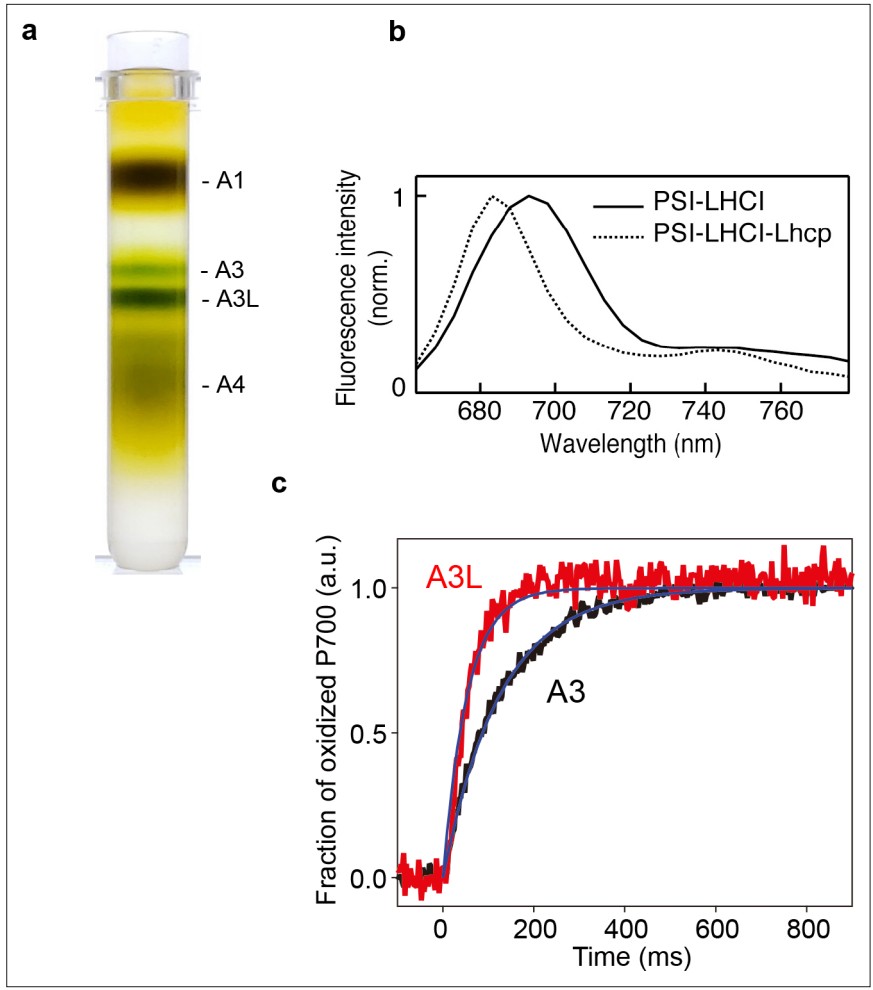

**Figure 1.** Characterization of A3 and A3L fractions. (**a**) Sucrose density gradient showing four major bands corresponding to free LHCs (A1), PSI-LHCI supercomplex (A3), PSI-LHCI-Lhcp supercomplex and PSII-Lhcp supercomplex (A3L), and other complexes (A4). (**b**) 77 K steady-state fluorescence spectrum of PSI-LHCI (solid line) and PSI-LHCI-Lhcp (dotted line). (**c**) PSI light-harvesting capabilities in the A3 and A3L fractions. Light-induced P700 oxidation kinetics of PSI were measured in A3 and A3L fractions under 28 µmol photon $m^{-2}$ $s^{-1}$. The fraction of P700 oxidation was derived from $\Delta(A_{820}-A_{870})$. *Solid* lines and *shaded* area represent averages of five (for A3) or ten (for A3L) technical replicates and SD, respectively. *Blue* lines represent fitting curves by mono-exponential functions. Data are representative of two biologically independent experiments. See another set of data in *Figure 1—figure supplement 3*.

The online version of this article includes the following source data and figure supplement(s) for figure 1:

**Source data 1.** Quantitative data for *Figure 1c*.

**Figure supplement 1.** Fractionation and characterization of the supercomplex samples from the *O. tauri* cells grown in the low light (50 µmol photon $m^{-2}$ $s^{-1}$).

**Figure supplement 1—source data 1.** Raw data for *Figure 1—figure supplement 1*.

**Figure supplement 2.** Fluorescence decay-associated spectra (FDAS) at 77 K (e.g., 405 nm, 4 µg Chl $mL^{-1}$).

**Figure supplement 3.** Photosystem I (PSI) light-harvesting capabilities in the A3 and A3L fractions.

**Figure supplement 3—source data 1.** Quantitative data for *Figure 1—figure supplement 3*.

**Figure supplement 4.** Structural analysis of A3 and A3L fractions by negative staining EM.

made up of <10 and 65 ps components (*Figure 1—figure supplement 2b*), which correspond to the total trapping time around P700, including the energy transfer between bulk Chl and P700 and the trapping at P700 (*Mimuro et al., 2010*). The positive peak in the fastest lifetime component observed in A3 was similar to that previously observed in the PSI core of cyanobacteria, which reflects the fast

**Table 1.** Pigment composition in the A3L fraction as revealed by UPLC analysis. Mean (±STD, n=3).

| Pigment | Molar ratio (Chl a=100) |
|---|---|
| Chl b | 36.2 (±0.1) |
| β-Carotene | 13.4 (±0.5) |
| Mdp | 3.8 (±0.1) |
| Uriolide | 4.9 (±0.1) |
| Prx | 11.8 (±0.1) |
| Nex | 3.6 (±0.1) |
| Vio | 3.3 (±0.1) |
| Micromonal | 2.3 (±0.1) |
| Dlt | 2.2 (±0.1) |

energy transfer process to P700 from the Chl near P700 and the subsequent fast charge separation. In A3L, however, the fastest lifetime component showed different shapes from that in A3 and it is also implying PSI-LHCI-Lhcp supercomplex in A3L. Because A3L has larger light-harvesting antennae, there are more Chls further away from P700 (e.g., those bound to Lhcp) than A3. Therefore, the long-range energy transfer process from the antennae to P700 becomes dominant in the observation, and the subsequent early trapping process could be masked. The slight increase in the lifetimes of A3L compared to those of A3 (<10–20 ps, 65–90 ps) likely reflects the presence of larger antennae in A3L. The light-induced oxidation kinetics of P700 indeed showed that the PSI antenna size in A3L was larger than that of A3 (*Figure 1c*, *Figure 1—figure supplement 3*). Notably, the 77 K fluorescence spectra of PSI-LHCI and PSI-LHCI-Lhcp exhibit a peak at 690 and 680 nm, respectively (*Figure 1b*), and there are no distinctive far-red peaks as reported previously (*Swingley et al., 2010*). The blue-shifted fluorescence was more prominent in PSI-LHCI-Lhcp fraction. These results and the EM analysis of the negatively stained particles (*Figure 1—figure supplement 4*, *Table 2*) suggest that A3 almost exclusively consists of PSI-LHCI supercomplex, while A3L was mainly composed of PSI-LHCI-Lhcp supercomplex. We thus proceeded to solve the cryo-EM structure of the large PSI-LHCI-Lhcp supercomplex in A3L in order to reveal its detailed architecture.

## Supramolecular assembly of *Ot*Lhcp trimers (Trimers) with PSI-LHCI complex

Through the single-particle cryo-EM method, the structure of *Ot*PSI-LHCI-Lhcp supercomplex is solved at an overall resolution of 2.94 Å and the local regions of three Lhcp trimers (namely Trimers 1–3) are refined to 2.9–3.5 Å (*Figure 2—figure supplement 1*, *Table 3*). As shown in *Figure 2*, the *Ot*PSI-LHCI-Lhcp supercomplex consists of a central PSI monomer encircled by six LHCIs and three Trimers. *Ot*PSI contains two large core subunits (PsaA and PsaB), three subunits on the stromal surface (PsaC, PsaD, and PsaE) (*Figure 2a–c*), nine small membrane-embedded subunits (PsaF, PsaG, PsaH, PsaI, PsaJ, PsaK, PsaL, PsaM, PsaO), one subunit on the lumenal surface (PsaN) (*Figure 2d–f*). In the supercomplex, a total of 314 Chls (245 Chl a, 60 Chl b, 9 magnesium 2,4-divinylpheoporphyrin $a_5$ monomethyl ester [Mdp], 104 carotenoids, 2 phylloquinones, and 3 $Fe_4S_4$ clusters have been located [*Table 4*]). The densities for representative Chl, carotenoid, and lipid molecules as well as two small subunits (PsaM and PsaN) are shown in *Figure 2—figure supplement 2*. To our best knowledge, the structure of *Ot*PSI has the largest number of subunits among the structures of PSI known so far (including those from plants, green algae, diatom, red algae, and cyanobacteria, *Figure 3*).

Four Lhca (Lhca1-4) complexes are associated with the PSI core at the side of PsaG- F-J-K (*Figure 2a and d*), similar to those found in plant PSI-LHCI complexes (*Mazor et al., 2017*; *Qin et al., 2015*; *Figure 3*). On the other side formed

**Table 2.** Two-dimensional classification of the photosystem I (PSI) particles in the A3 and A3L fractions.

The RELION 2.1 package (*Kimanius et al., 2016*) was used for automated particle picking of particles and Two-dimensional (2D) classification into 50 classes as previously described (*Watanabe et al., 2019*). Classes of poor quality due to aggregation, contamination, micrograph edge, or extreme proximity were discarded. 2D classification was performed on the PSI particles and manually assigned to respective small and large PSI supercomplexes.

| Fraction | A3 | A3L |
|---|---|---|
| Small PSI (PSI-LHCI) | 2675 (100%) | 1030 (30%) |
| Large PSI (PSI-LHCI-Lhcp) | | 2437 (70%) |
| Total PSI particles | 2675 | 3467 |

**Table 3.** Statistics of structural analysis of the OtPSI-LHCI-Lhcp supercomplex.

| | *OtPSI-LHCI-Lhcp* |
|---|---|
| **Data collection and processing** | |
| Magnification | 130,000 |
| Voltage (kV) | 300 |
| Electron exposure (e⁻ Å⁻²) | 60 |
| Defocus range (μm) | from –1.8 to –2.2 |
| Pixel size (Å) | 1.04 |
| Symmetry imposed | C1 |
| Initial particle images (no.) | 5,288,217 |
| Final particle images (no.) | 80,366 |
| Map resolution (Å) | 2.94 |
| FSC threshold | 0.143 |
| Map resolution range (Å) | 2.3–4.3 |
| | |
| **Refinement** | |
| Initial model used (PDB code) | 5ZJI, 7D0J |
| Model resolution (Å) | 3.0 |
| FSC threshold | 0.5 |
| Map sharpening B factor (Å²) | –110.961 |
| Model composition | |
| Nonhydrogen atoms | 64,836 |
| Protein residues | 5632 |
| Ligands | 450 |
| B factor (Å²) | |
| Protein | 56.82 |
| Ligand | 56.55 |
| R.m.s. deviations | |
| Bond lengths (Å) | 0.011 |
| Bond angles (°) | 1.79 |
| Validation | |
| MolProbity score | 1.51 |
| Clash score | 5.41 |
| Poor rotamers (%) | 0 |
| Ramachandran plot | |
| Favored (%) | 96.6 |
| Allowed (%) | 3.37 |
| Outliers (%) | 0.04 |

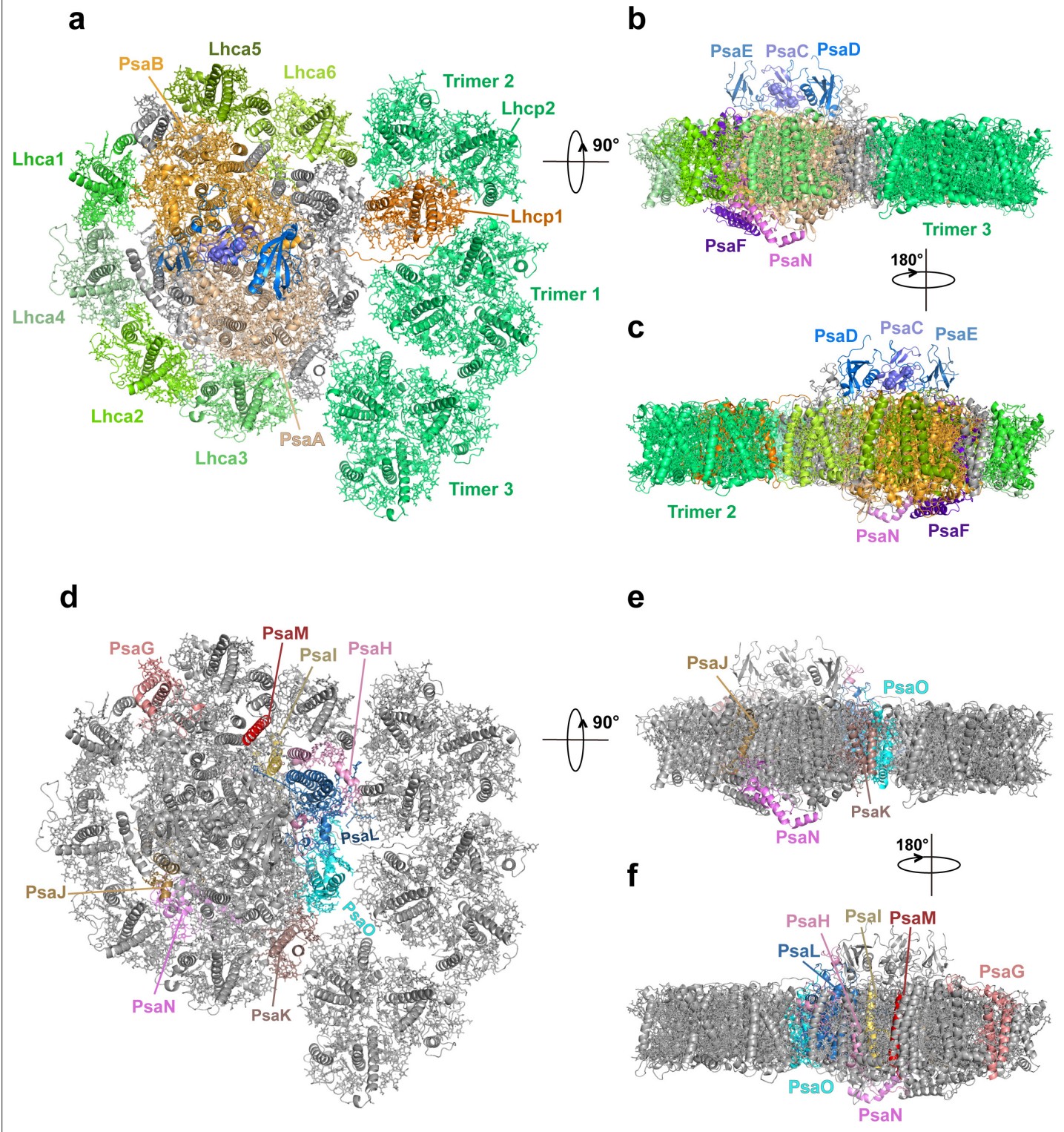

**Figure 2.** Overall architecture of *Ot*PSI-LHCI-Lhcp supercomplex. (**a**) Top view of the supercomplex from stromal side along membrane normal. (**b**) and (**c**) Two different side views of the supercomplex along the membrane plane. While the protein backbones are shown as cartoon models, the pigment and lipid molecules are presented as stick models. The iron-sulfur clusters are presented as sphere models. The bulk region of photosystem I (PSI) core is colored in wheat, while Lhca1-6, Lhcp1, and Lhcp2, PsaC, PsaD, PsaE, PsaF, and PsaN are highlighted in different colors. The remaining small subunits are in silver. (**d–f**) The small subunits at the interfaces between PSI core and LHCI/Lhcp complexes. The viewing angles are the same as (**a–c**), whereas the color codes are different. The interfacial small subunits are highlighted in various colors, while the PSI core and LHCI/Lhcp complexes are in silver.

*Figure 2 continued on next page*

*Figure 2 continued*

The online version of this article includes the following figure supplement(s) for figure 2:

**Figure supplement 1.** Cryo-EM data collection, processing, refinement, and validation statistics of *Ot*PSI-LHCI-Lhcp structures.

**Figure supplement 2.** Cryo-EM densities of various cofactors and protein subunits found in the PSI-LHCI-Lhcp supercomplex of *O. tauri*.

**Figure supplement 3.** The detailed local cryo-EM map features for identification of *Ot*Lhcp1 and *Ot*Lhcp2.

by PsaG-I-M-H, an Lhca5-Lhca6 heterodimer is associated with the PSI core (*Figure 2a and d*). The location of Lhca5-Lhca6 heterodimer overlaps with that of Lhca9-Lhca2 heterodimer associated with *C. reinhardtii* PSI (*Su et al., 2019*; *Suga et al., 2019*) or Lhcr2-Lhcr1 heterodimer associated with *Cyanidioschyzon merolae* PSI (*Pi et al., 2018*), whereas the corresponding sites are vacant in plant PSI structures (*Mazor et al., 2017*; *Pan et al., 2018*; *Qin et al., 2015*; *Yan et al., 2021*). While the fourth transmembrane helix of *Cr*Lhca2 or Lhca5 from *Dunaliella salina* (also known as transmembrane helix F/TMF or TMH4), located at the dimerization interface of the Lhca2/Lhca9 or Lhca5-Lhca6 heterodimer, was previously proposed to replace the role of PsaM in mediating assembly of the LHCI heterodimer with PSI (*Caspy et al., 2020*; *Suga et al., 2019*), the fourth helix of *Ot*Lhca6 and PsaM coexist in *O. tauri*. As PsaM is also present in moss (*Gorski et al., 2022*) and cyanobacteria (*Jordan et al., 2001*), where LHCI heterodimer is absent at this position, the role and the origin of the fourth helix of the Lhca proteins might not be directly related to PsaM and need to be revisited.

Previously, it was found that plant and *C. reinhardtii* PSI complexes contain some red-form Chls, absorbing photons at energy levels below that of the primary donor and mainly located in the LHCI complexes (*Croce and van Amerongen, 2013*). Unlike plant and *C. reinhardtii* PSI, *Ot*PSI-LHCI-Lhcp supercomplex sample does not exhibit far-red peaks in the 77 K fluorescence spectra (*Figure 1b*). In the *Ot*PSI-LHCI, the Chl *a*603-*a*609 pairs of Lhca1-4 (corresponding to the red-form Chls found in plant and *C. reinhardtii* Lhca1-4) are separated at larger distances, while those in Lhca5 and Lhca6 are similar to the ones in Lhca9 and Lhca2 from *C. reinhardtii* (*Figure 4*). Moreover, the axial ligands of Chl *a*603 in Lhca1-4 from *O. tauri* are all His residues instead of Asn (*Figure 4b–e*). Mutation of Asn to His for the ligand of Chl *a*603 in plant Lhca3 and Lhca4 led to absence of red spectral forms, while substitution of Asn for His in Lhca1 caused red shift of the fluorescence emission (*Morosinotto et al., 2003*). As His has longer side chain than Asn, the position of Chl *a*603 in *Ot*Lhca1-4 is located farther away from the protein backbone (in comparison with those from plant and *C. reinhardtii* LHCIs) so that the distance between Chl *a*603 and Chl *a*609 becomes larger and their excitonic coupling strength might be reduced as a result. The axial ligands of Chl *a*603 in Lhca5-Lhca6 dimer from *O. tauri* are both Asn residues same as those in the Lhca9-Lhca2 dimer on a similar location in *C. reinhardtii* (*Figure 4b–e*). Although Chl *a*603-*a*609 pairs of Lhca9 and Lhca2 were proposed to be responsible for the red spectral forms in *C. reinhardtii* (*Mozzo et al., 2010*), those in *Ot*Lhca5 and *Ot*Lhca6 with similar configuration do not cause the red spectral forms (*Figure 4f and g*), indicating that the presence of Asn residues at the axial ligand site is not sufficient to cause the spectral red form of Chl *a*603/*a*609. The distinct spectroscopic features of *Ot*PSI-LHCI-Lhcp supercomplex might also be related to the local environments around chlorophyll molecules. For instance, Tyr69 and Tyr75 in *Ot*Lhca5 and *Ot*Lhca6 forms van der Waals contact with Chl $a609_{Lhca5}$ and Chl $a609_{Lhca6}$, respectively (*Figure 4f and g*). In comparison, these residues are replaced by tryptophan residues (Trp65) in *C. reinhardtii* Lhca9 and Lhca2. Previously, it was reported that tryptophan residues located nearby chlorophyll molecules may induce deformation of the tetrapyrrole macrocycle and cause red shift in the $Qy$ absorption bands of chlorophylls (*Bednarczyk et al., 2016*). Mutation of a tryptophan residue (in van der Waals contacts with a bacteriochlorophyll) to a Tyr or Phe residue caused a blue shift of the $Qy$ absorption peak in the core light-harvesting complex of *Rhodobacter sphaeroids* (*Sturgis et al., 1997*). Therefore, the occurrence of Tyr69 or Tyr75 residues (instead of Trp residues) around Chl $a609_{Lhca5}$ or Chl $a609_{Lhca6}$ may account (at least in part) for the lack of red spectral forms in the *Ot*PSI-LHCI-Lhcp supercomplex.

Besides the six Lhca proteins, three Lhcp Trimers (Trimers 1–3) bind to the PSI-LHCI complex on the third side along the surfaces of Lhca6, PsaH, PsaL, PsaO, and PsaK subunits (*Figure 2a and d*). As a result, the PSI core is enclosed by an irregular annular belt formed by the LHCI, Lhcp complexes, PsaG and PsaK (*Figure 2a and d*). This side of the PSI core was partly filled by one LHC trimer or two LHC trimers in higher plants or green algae when they are under state 2 conditions (*Huang et al., 2021*; *Pan et al., 2018*; *Pan et al., 2021*).

**Table 4.** Summarization of the components in the final structural model of the *Ot*PSI-LHCI-Lhcp supercomplex.

| Subunit | Number of amino acid residues traced | Chlorophylls | Carotenoids | Lipids | Others |
|---|---|---|---|---|---|
| PsaA | 742 | 44 Chl *a*<br>1 Cl0 | 6 BCR | 2 PG<br>2 MGDG<br>1 DGDG | 1 PQN<br>1 Fe₄S₄ |
| PsaB | 732 | 40 Chl *a* | 8 BCR | 1 PG<br>1 DGDG | 1 PQN |
| PsaC | 80 | | | | 2 Fe₄S₄ |
| PsaD | 143 | | | | |
| PsaE | 62 | | | | |
| PsaF | 165 | 3 Chl *a* | 1 BCR | | |
| PsaG | 95 | 3 Chl *a* | 1 BCR | | |
| PsaH | 96 | 3 Chl *a* | | 1 SQDG | |
| PsaI | 35 | | 1 BCR | | |
| PsaJ | 41 | 1 Chl *a* | 1 BCR | 2 MGDG | |
| PsaK | 87 | 4 Chl *a* | 3 BCR | 1 PG | |
| PsaL | 158 | 5 Chl *a* | 4 BCR | 1 PG<br>1 MGDG | |
| PsaM | 31 | | | | |
| PsaN | 91 | 2 Chl *a* | | | |
| PsaO | 96 | 5 Chl *a* | 2 DLT | 2 MGDG | |
| Trimer 2 | 201(Lhcp2) | 8 Chl *a*<br>5 Chl *b*<br>1 DVP | 2 PRX<br>4 DLT<br>1 NEX | | |
| | 225(Lhcp1) | 8 Chl *a*<br>5 Chl *b*<br>1 DVP | 2 PRX<br>4 DLT<br>1 NEX | | |
| | 201(Lhcp2) | 8 Chl *a*<br>5 Chl *b*<br>1 DVP | 2 PRX<br>4 DLT<br>1 NEX | | |
| Trimer 1 | 202(Lhcp2) | 8 Chl *a*<br>5 Chl *b*<br>1 DVP | 2 PRX<br>4 DLT<br>1 NEX | | |
| | 201(Lhcp2) | 8 Chl *a*<br>5 Chl *b*<br>1 DVP | 2 PRX<br>4 DLT | | |
| | 201(Lhcp2) | 8 Chl *a*<br>5 Chl *b*<br>1 DVP | 1 BCR<br>2 PRX<br>3 DLT | | |
| Trimer 3 | 201(Lhcp2) | 8 Chl *a*<br>5 Chl *b*<br>1 DVP | 1 BCR<br>2 PRX<br>3 DLT | | |
| | 200(Lhcp2) | 8 Chl *a*<br>5 Chl *b*<br>1 DVP | 2 PRX<br>4 DLT<br>1 NEX | | |
| | 200(Lhcp2) | 8 Chl *a*<br>5 Chl *b*<br>1 DVP | 1 PRX<br>4 DLT | | |
| Lhca1 | 195 | 9 Chl *a*<br>2 Chl *b* | 1 XAT<br>1 PRX | | |

*Table 4 continued on next page*

*Table 4 continued*

| Subunit | Number of amino acid residues traced | Chlorophylls | Carotenoids | Lipids | Others |
|---|---|---|---|---|---|
| Lhca2 | 205 | 11 Chl *a* 4 Chl *b* | 1 XAT 1 BCR 1 DLT 1 PRX | 1 PG 1 MGDG | |
| Lhca3 | 227 | 13 Chl *a* 1 Chl *b* | 1 XAT 3 BCR 1 PRX | 3 PG 1 MGDG | |
| Lhca4 | 205 | 11 Chl *a* 4 Chl *b* | 1 XAT 1 BCR 1 PRX | | |
| Lhca5 | 185 | 9 Chl *a* 1 Chl *b* | 1 XAT 1 PRX | 1 MGDG | |
| Lhca6 | 218 | 10 Chl *a* 3 Chl *b* | 1 XAT 1 PRX | 1 PG 1 MGDG 1 SQDG | |
| Chain Y | | | | | 4 $H_2O$ |

Among the three trimers associated with *Ot*PSI, Trimers 1 and 3 are both (Lhcp2)$_3$ homotrimers, whereas Trimer 2 is a Lhcp1(Lhcp2)$_2$ heterotrimer. The detailed cryo-EM map features for identification of Lhcp1 and Lhcp2 are shown in *Figure 2—figure supplement 3*. Trimers 1 and 3 bind to PSI on the PsaO and PsaK sides, respectively, while Trimer 2 assembles with PSI on the PsaL-PsaH side through Lhcp1 subunit and interacts with Lhca6 through an Lhcp2 subunit. As Trimer 1 is sandwiched between Trimers 2 and 3, it forms close contacts with both Trimers 2 and 3, and is related with them through pseudo-C2 symmetry axes at their interfaces. In *C. reinhardtii*, two LHCII trimers associate with PSI in state 2 (*Huang et al., 2021*; *Pan et al., 2021*), whereas one LHCII trimer is located at the peripheral region of *Zea mays* PSI (*Zm*PSI) in state 2 (*Pan et al., 2018*; *Figure 3*). While the binding sites of Trimers 1 and 2 partially overlaps with those of LHCII-1 and LHCII-2 trimers from *Cr*PSI-LHCI-LHCII supercomplexes respectively, they do not superpose well with each other (*Figure 5*). When the PSI core regions are aligned, *O. tauri* Trimer 1 appears to be rotated by 57 degrees and shifted by 16.1 Å in relation to the position of *C. reinhardtii* LHCII-1. Trimer 2 is rotated by 52 degrees in relation to the position of *C. reinhardtii* LHCII-2. The LHCII trimer associated with *Zm*PSI binds to a position between Lhcp Trimers 1 and 3.

Trimer 1 is located nearby PsaO and the closest interfacial distances between them are 7.0 Å or larger (*Figure 6a*). Although Trimer 1 does not form direct interactions with PsaO, we cannot rule out the presence of unresolved lipid molecules that may mediate the interactions. On the other hand, Trimers 2 and 3 form close and direct interactions with PsaL-PsaH and PsaK, respectively (*Figure 6b and c*). Trimer 2 binds to PSI on three different sites (*Figure 6d–f*). At site 1, Lhcp1$_{Trimer2}$ has its elongated N-terminal region (NTR) partially inserted into a surface pocket formed by PsaL and PsaH subunits (*Figure 6b*). The NTR of Lhcp1 contains an RRpT (pT, phosphorylated Thr residue) motif identical to those found in *Z. mays* pLhcb2 (*Pan et al., 2018*) and *C. reinhardtii* pLhcbM1 (*Pan et al., 2021*). The RRpT motif of Lhcp1 interacts with nearby amino acid residues through salt bridges and hydrogen bonds (*Figure 6d*), in a similar way as those of *Z. mays* pLhcb2 and *C. reinhardtii* pLhcbM1. Arg29 in Lhcp1 might be acetylated on its main-chain amine group as there is an extra density linked to it, and the van der Waals interactions between the putative acetyl group and nearby amino acid residues from PsaH serve to stabilize the association between Lhcp1 and the PSI core (*Figure 6d*). Consistently, spinach Lhcb proteins are also acetylated at their amino-terminal arginine and phosphorylated on threonine/serine in the third position (*Michel et al., 1991*). Sites 2 and 3 are located in the membrane-embedded regions on the stromal and lumenal sides, respectively (*Figure 6b*). On these two sites, Lhcp1 forms hydrogen bonds and van der Waals interactions with nearby amino acid residues and pigment molecules from PsaH and PsaL subunits (*Figure 6e and f*). Meanwhile, Trimer 3 associates with PsaK mainly through van der Waals interactions on the stromal and lumenal sides (*Figure 6g*).

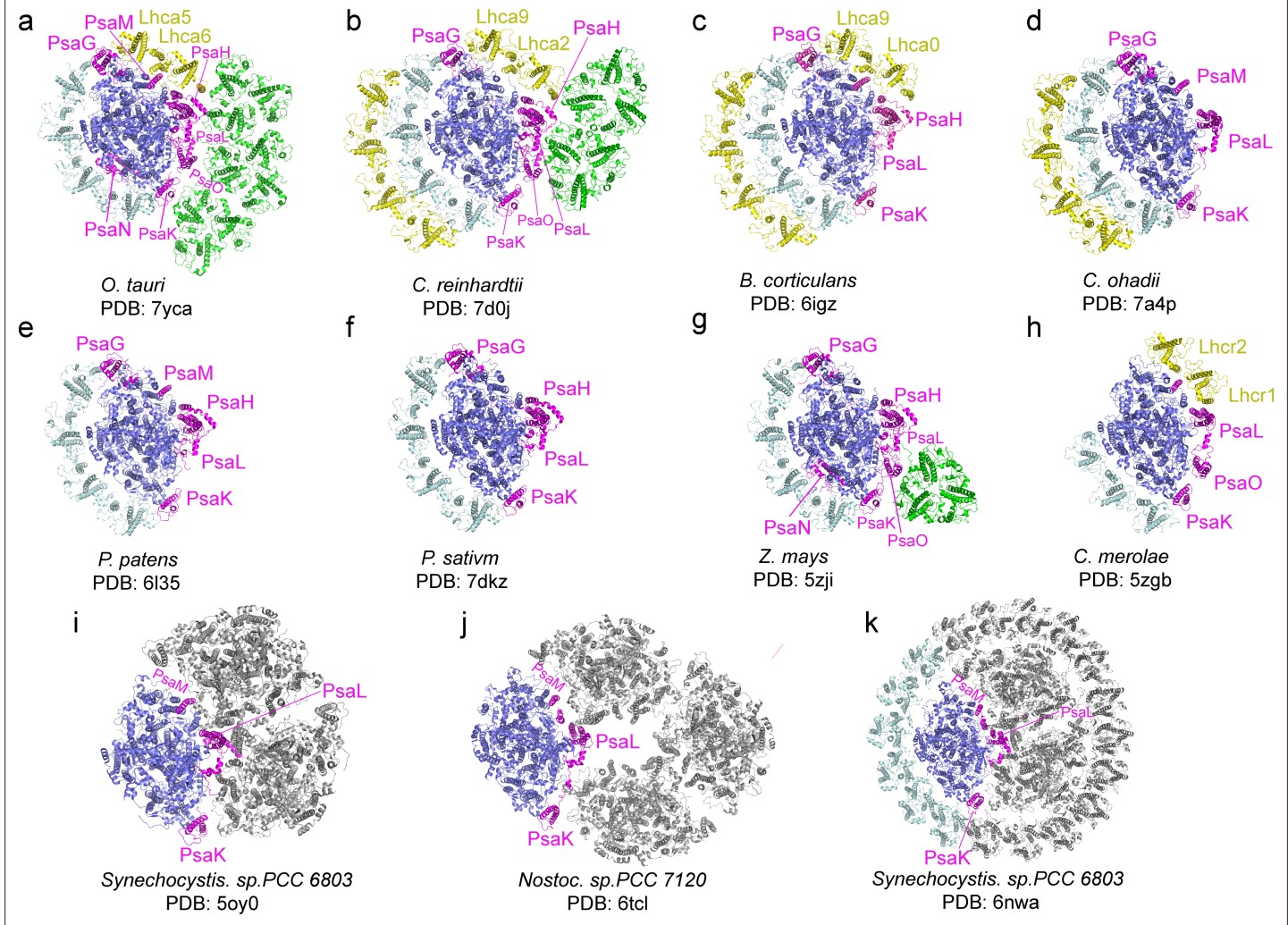

**Figure 3.** Comparison of *O.tauri* PSI-LHCI-Lhcp supercomplex with the photosystem I (PSI) supercomplexes from other organisms. (**a**) *Ot*PSI-LHCI-Lhcp supercomplex. (**b–k**) Structures of PSI from different species or at different states. *Blue*, large PSI core subunits; *magenta*, small PSI core subunits; *yellow*, Lhca5-Lhca6/Lhca2-Lhca9/Lhca0-Lhca9/Lhc1-Lhc2 and the outer belt of LHCI; *cyan*, Lhca1-Lhca2-Lhca3-Lhca4 and the inner belt of antenna complexes around PSI from other species; *green*, Lhcp or LHCII trimers; *gray*, symmetry-related units in trimeric or tetrameric PSI.

## Characteristic features of Lhcp1 and Lhcp2 monomers

As an ancient member of the green lineage, *O. tauri* belongs to Mamiellales of Prasinophyceae at the basal position of the green lineage. While *Ot*Lhcp1 and *Ot*Lhcp2 are evolutionarily related to plant Lhcbs and *Cr*LhcbMs (***Six et al., 2005***), they only share low sequence identity, e.g., 32–37% sequence identity with *Zm*Lhcb2 or *Cr*LhcbM1. The apoproteins of *Ot*Lhcp1 and *Ot*Lhcp2 adopt a classical fold of Lhc family with three transmembrane helices (A, B, and C) and three short amphipathic helices on the lumenal side (D, E, and F) (***Figure 7a and b*** and ***Figure 7—figure supplement 1***). While the structure of *Ot*Lhcp1 highly resembles that of *Ot*Lhcp2, it differs from those of *Zm*Lhcb2 and *Cr*LhcbM1 in the NTR, EC loop, AC loop, and CTR (***Figure 7—figure supplement 2a–c***). Besides, helices B, A, and C of *Ot*Lhcp1/2 are slightly shorter than the corresponding ones in *Zm*Lhcb2 or *Cr*LhcbM1 (***Figure 7—figure supplement 2b, c***).

In terms of pigment composition, *Ot*Lhcp1 and *Ot*Lhcp2 each contain 14 Chl molecules (8 Chl *a*, 5 Chl *b*, and 1 Mdp) and 7 carotenoid molecules (4 Dlt, 2 Prx, and 1 Nex) (***Figure 7a and b***). The Chl *b/a* ratio of the structural model is 0.625, close to the previously reported value of 0.736 for the Lhcp2 preparation (***Swingley et al., 2010***). While the number of Prx molecules found in the structure matches the previous prediction, four (instead of one) Dlt molecules are more than expected (***Swingley et al., 2010***). As *O. tauri* OTH95 species thrives in lagoons and shallow area of the ocean

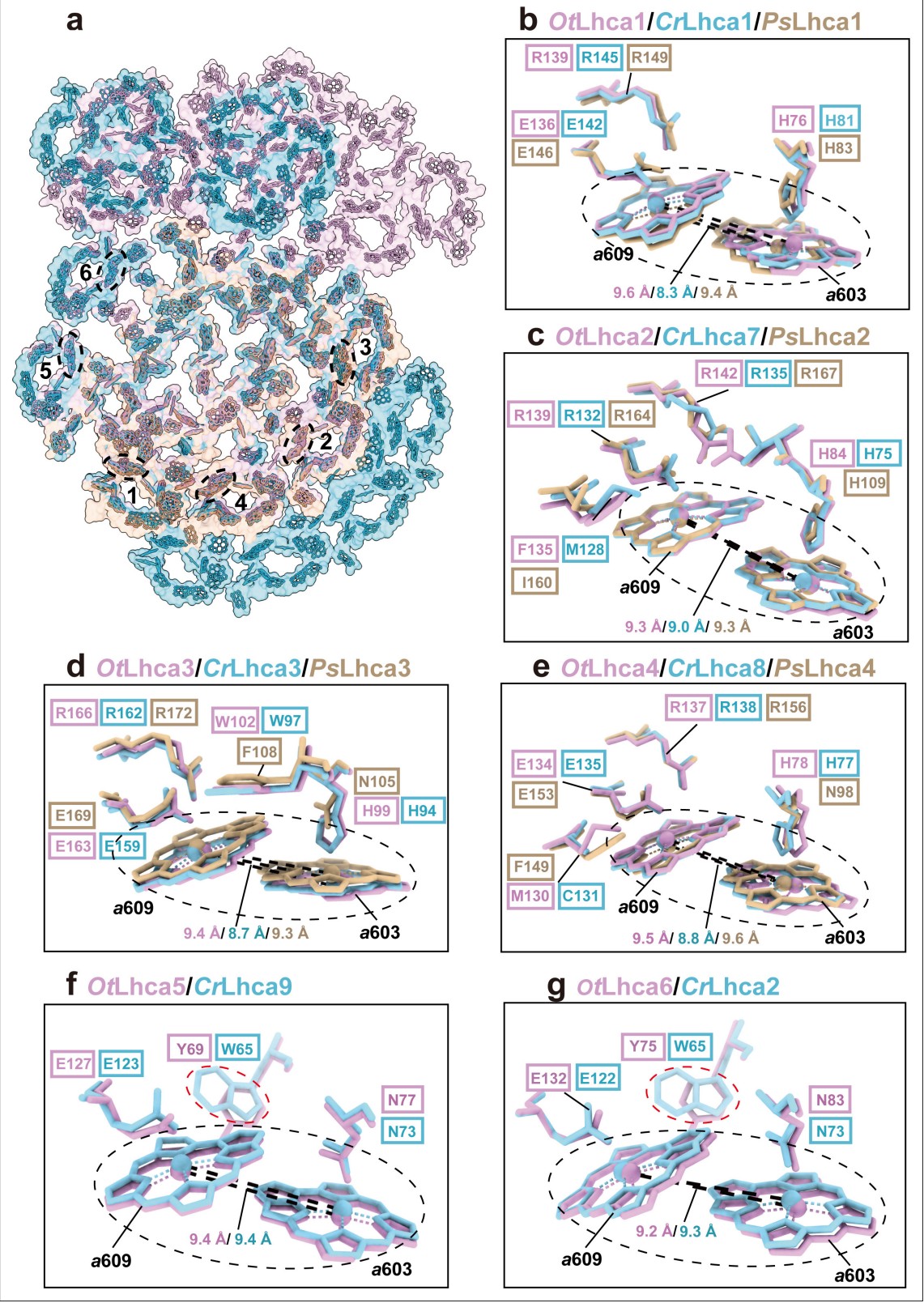

**Figure 4.** Comparison of Chl *a*603-*a*609 dimers in Lhca complexes among *O. tauri*, *C. reinhardtii*, and *P. sativum*. (**a**) Overall arrangement of the Chl *a*603-*a*609 dimers in six Lhca complexes of *O. tauri*, *C. reinhardtii*, and *P. sativum*. The chlorophylls in the photosystem I (PSI) supercomplexes are presented in surface models and those of Lhca complexes are superposed with stick models. Color code: *pink, O. tauri; golden, P. sativum; light blue, C. reinhardtii*. The dark labels (1–6) indicate the Lhca1-Lhca6 subunits and the dashed ovals label the locations of the *a*603-*a*609 dimer in each LHCI.

*Figure 4 continued on next page*

*Figure 4 continued*

(**b–g**) Comparison of the Chl *a*603-*a*609 dimers and their local environments in six different Lhca subunits from the three different species. Note that the axial ligands of Chl *a*603 in Lhca1-4 from *O. tauri* are all His, while the Asn in the 603 site of plant Lhca3 and Lhca4 are crucial for the formation of the red-most form chlorophyll. The Lhca5/Lhca9 and Lhca6/Lhca2 are only present in *O. tauri* and *C. reinhardtii* but absent in *P. sativum*. The Chl *a*603-*a*609 dimers are indicated in the black dashed ovals and the key amino acid residues around the two chlorophylls are shown as stick models. The number labeled nearby the black dashed lines indicate the Mg-Mg distances between Chl *a*603-*a*609 dimer in Lhca complexes from *O. tauri*, *C. reinhardtii,* and *P. sativum*. In (**f**) and (**g**), the red dashed ovals indicate the Tyr/Trp residues around the Chl *a*609 molecules from *Ot*Lhca5/*Cr*Lhca9 and *Ot*Lhca6/*Cr*Lhca2.

frequently challenged by high light with an intensity up to 2000 µmol photons m$^{-2}$ s$^{-1}$ (*Courties et al., 1994*), incorporation of a large number of carotenoid molecules in *Ot*Lhcp1 and *Ot*Lhcp2 may help to protect the algae from the damaging effect of harmful excess excitation energy, by quenching triplet-state Chl, scavenging toxic oxygen species or quenching of the singlet excited-state Chl (*Bassi and Dall'Osto, 2021*; *Young, 1991*).

The Chl molecules in *Ot*Lhcp1/*Ot*Lhcp2 are arranged into two membrane-embedded layers close to the stromal and lumenal surfaces, respectively (*Figure 7c and d*). The stromal and lumenal layer each has seven Chl molecules. While the stromal layer (five Chl *a*, one Chl *b*, and one Mdp) contains more Chl *a* than Chl *b*, the lumenal layer (three Chl *a* and four Chl *b*) harbors more Chl *b* than Chl *a*. In the stromal layer, six Chls (*a*602, *a*603, *b*608, *a*610, *a*611, and *a*612) can find their counterparts of the same identity in plant or *C. reinhardtii* LHCII (*Figure 7—figure supplement 2d–f*). The 609 site is occupied by an Mdp molecule in *Ot*Lhcp1/2 (*Figure 2—figure supplement 2*), and the free carboxyl group on the side chain of porphyrin ring IV in the Mdp609 molecule forms a salt bridge with Lys77 residue nearby. In comparison, plant LHCII has the 609 site occupied by a Chl *b* (*Liu et al., 2004*). The Chl *b*609 in LHCII has its C7-formyl group forming a hydrogen bond with Gln131 on the Lhcb1 protein, and the side chain amine group of Gln131 is crucial for selective binding of Chl *b*609 and *b*607 in LHCII (*Bassi et al., 1999*). Lhcp1 and Lhcp2 adopt a Glu residue in the position corresponding to Gln131 of the Lhcb1 protein (*Figure 8*). Similar change is also observed in CP29 or CP26 with lower Chl *b* content than LHCII, and Q131E mutation of Lhcb1 led to decrease of Chl *b* content and increase

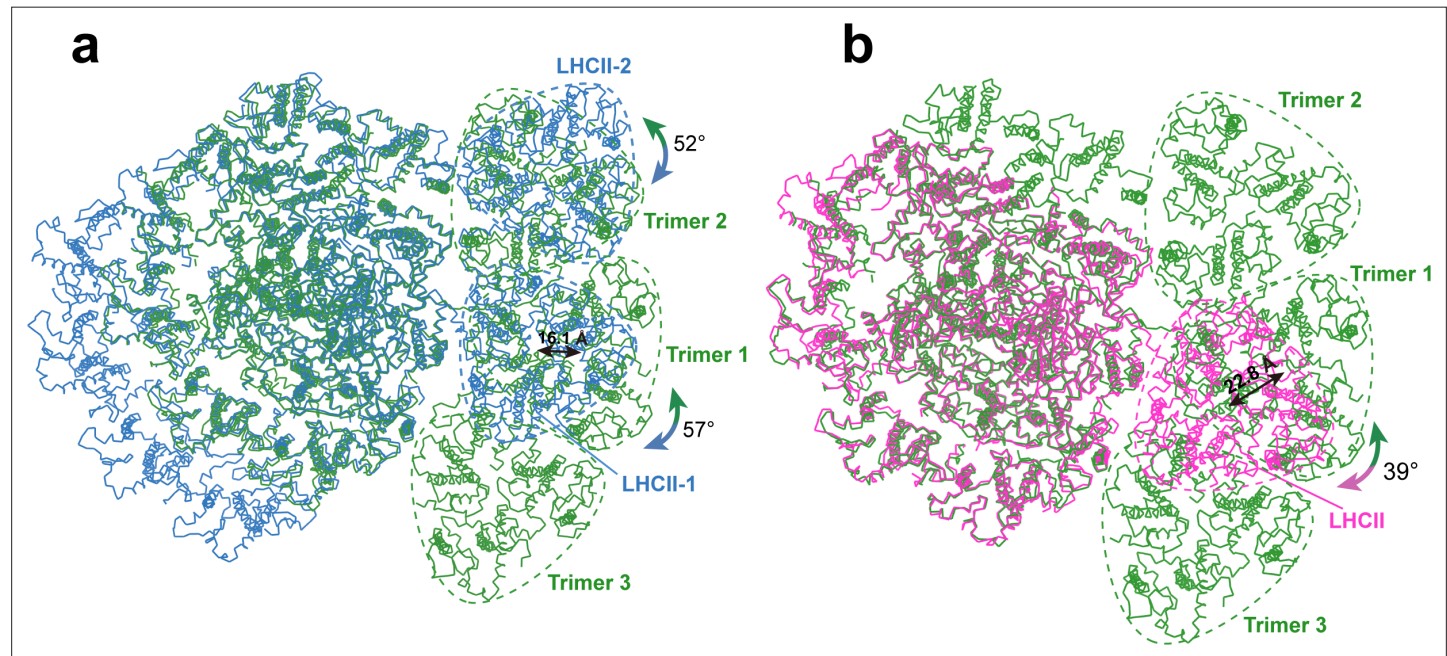

**Figure 5.** Comparing the binding sites of Trimers with those of LHCII trimers bound to *C. reinhardtii* and plant photosystem I (PSI). (**a and b**) The structure of *Ot*PSI-LHCI-Lhcp supercomplex is superposed with the PSI-LHCI-LHCII supercomplex from *C. reinhardtii* (PDB code: 7DZ7, **a**) or *Z. mays* (PDB code: 5ZJI, **b**). The three structures are superposed on the common PsaA subunits. The dash triangular rings outline the approximate boundaries of Trimers or LHCII trimers. Color code: *green*, *Ot*PSI-LHCI-Lhcp; *blue*, *Cr*PSI-LHCI-LHCII; *magenta*, *Zm*PSI-LHCI-LHCII. The double-headed arrows indicate the translational or rotational relationships between Lhcp trimers in *O. tauri* and the corresponding LHCII trimers in *C. reinhardtii* and in *Z. mays*. The number labeled nearby the arrows indicate the translation distances or rotation angles.

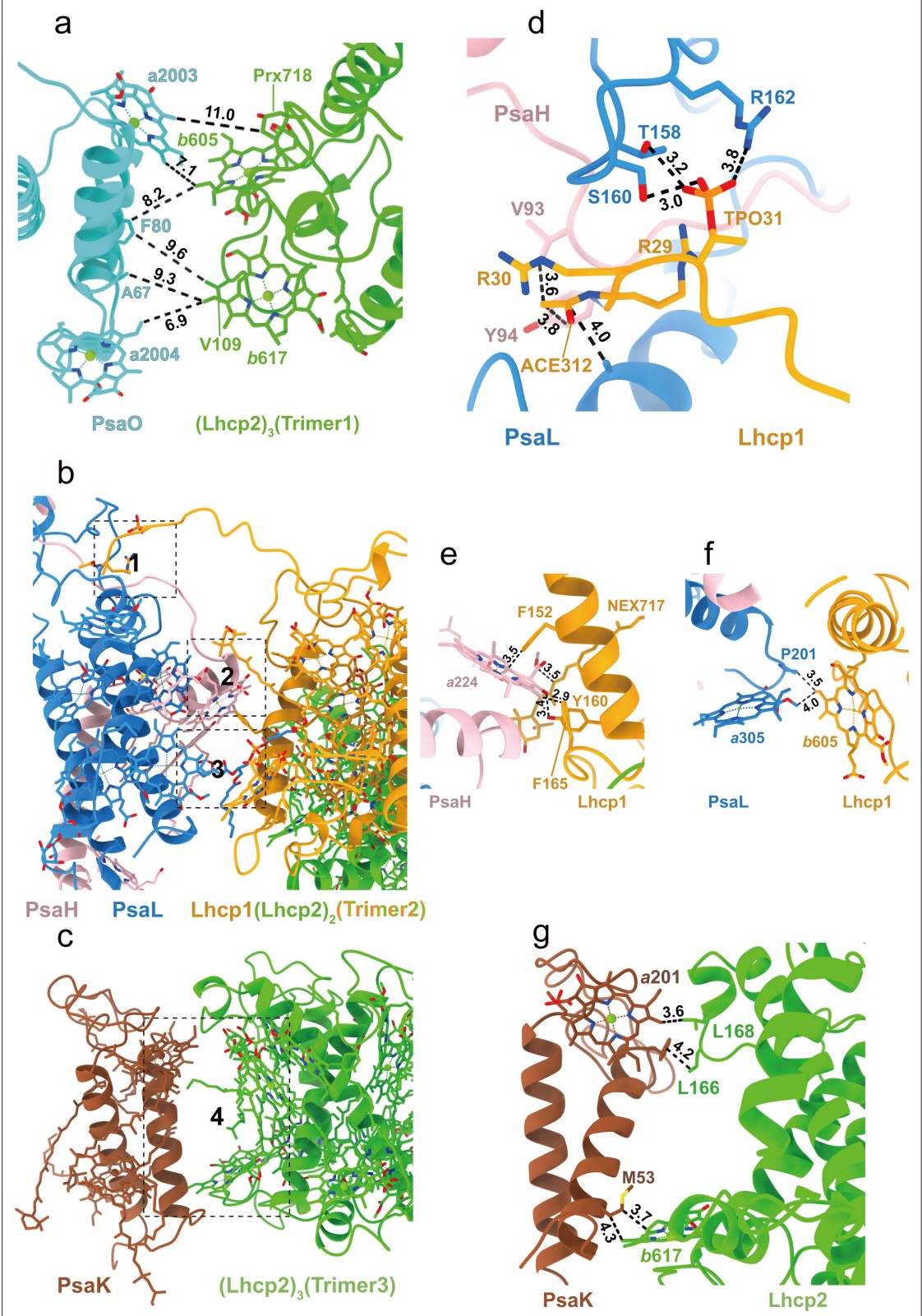

**Figure 6.** Interfaces between Trimers and photosystem I (PSI) subunits. (**a**) The interface between Trimer 1 and PsaO. (**b**) The interface between Trimer 2 and PsaL-PsaH. (**c**) The interface between Trimer 3 and PsaK. (**d–f**) The detailed interactions between Lhcp1 of Trimer 2 and PsaL/PsaH at sites 1–3 shown in **b**. (**g**) The detailed interactions between Lhcp2 of Trimer 3 and PsaK at site 4 shown in **c**. The numbers labeled nearby the dash lines are distances (Å) between two adjacent groups.

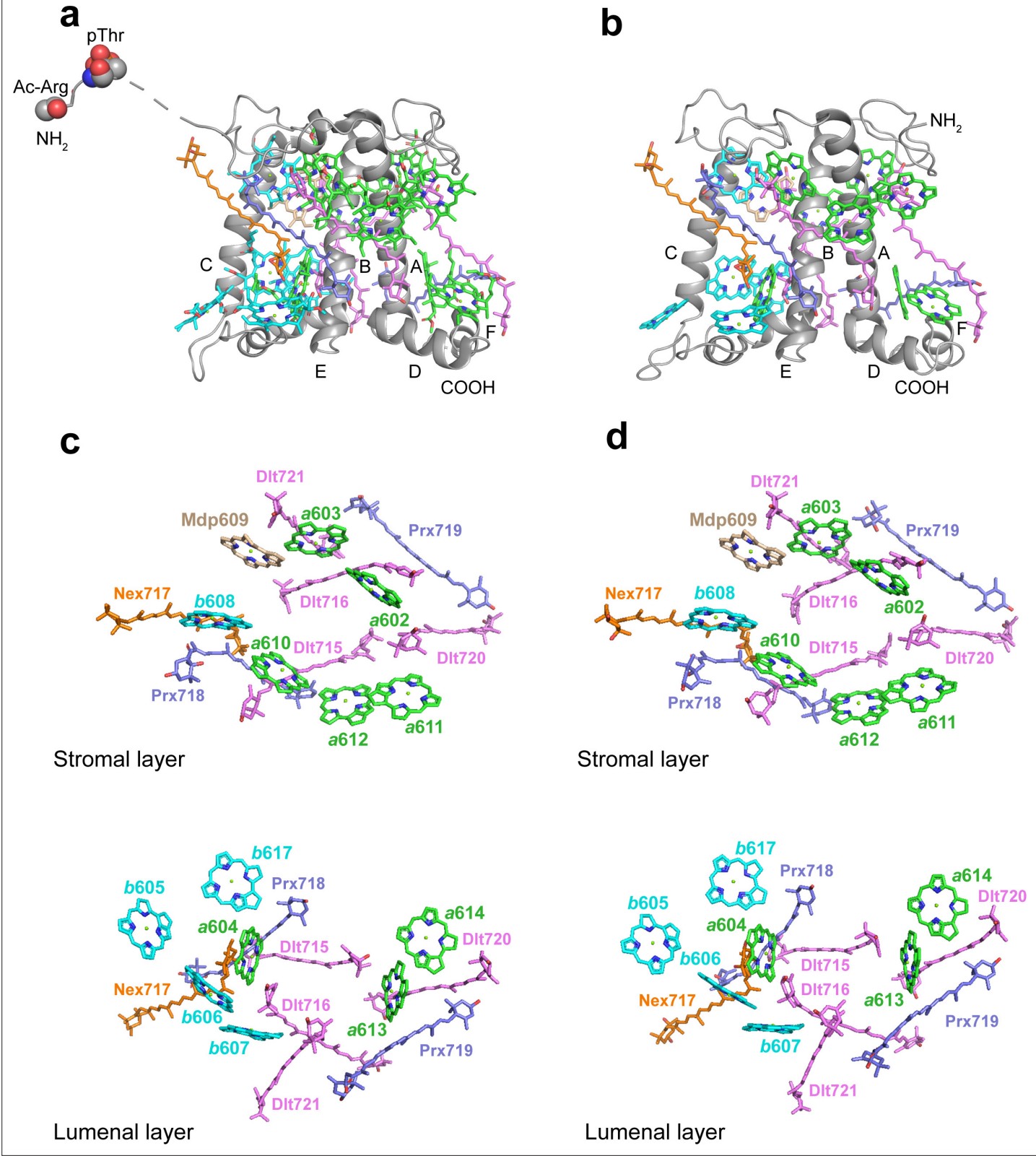

**Figure 7.** The structures and pigment compositions of *Ot*Lhcp1 and *Ot*Lhcp2. (**a and b**) Side views of *Ot*Lhcp1 (**a**) and *Ot*Lhcp2 (**b**) structures. The backbones of *Ot*Lhcp1/*Ot*Lhcp2 apoproteins are showed as cartoon models, while the pigments are presented as stick models. The phytyl chains of Chl molecules are omitted for clarity. The phosphorylated Thr residue and the acetylated Arg residue at the N-terminal region of Lhcp1 is highlighted as sphere models. A, B, and C indicate the three transmembrane helices in *Ot*Lhcp1/*Ot*Lhcp2 apoproteins, whereas D and E are the two amphipathic

*Figure 7 continued on next page*

*Figure 7 continued*

helices at the lumenal surface. Color codes: *gray*, Lhcp1 and Lhcp2 apoproteins; *green*, Chl *a*; *cyan*, Chl *b*; *magenta*, dihydrolutein/DLT; *purple*, prasinoxanthin/PRX; *orange*, neoxanthin/NEX. (**c and d**) The arrangement of pigment molecules in *Ot*Lhcp1 (**c**) and *Ot*Lhcp2 (**d**). For clarity, the apoproteins are not shown. In the upper row, only pigment molecules within the layer close to stromal surface are shown, while the lower row shows the pigment molecules within the layer close to the lumenal surface.

The online version of this article includes the following figure supplement(s) for figure 7:

**Figure supplement 1.** Cryo-EM densities of *Ot*Lhcp1 and *Ot*Lhcp2 proteins.

**Figure supplement 2.** Superposition of the *Ot*Lhcp1 structure with those of *Ot*Lhcp2 (**a, d**), *Zm*Lhcb1 (**b, e**), and *Cr*LhcbM1 (**c, f**).

of Chl *a* content (**Bassi et al., 1999**). As the side chain hydroxyl group of Glu150/Glu129 residue in *Ot*Lhcp1/2 forms a hydrogen bond with Chl *b*607, the 609 site is occupied by an Mdp molecule with a $C^7$-methyl group (instead of a Chl *b*) due to lack of hydrogen bond donor. Moreover, Chl *b*601 found in plant and *C. reinhardtii* LHCII is absent in *Ot*Lhcp1/2, mainly because the head group of a carotenoid molecule (Dlt720) occupies part of its binding site (**Figure 7—figure supplement 2e, f**) and the NTR adopts a conformation different from those of LHCII apoproteins (**Figure 7—figure supplement 2b, c**).

In the lumenal layer, six of the seven Chls (*a*604, *b*605, *b*606, *b*607, *a*613, and *a*614) can find their counterparts in plant LHCII. A previously unobserved Chl (Chl *b*617, absent in plant or *C. reinhardtii* LHCII) is found in the EC loop region of *Ot*Lhcp1/2 (**Figure 7c and d**, **Figure 7—figure supplement 2d**) and is likely coordinated by the backbone carbonyl of a proline residue conserved in both Lhcp1 and Lhcp2, but not in plant Lhcb2 or *Cr*LhcbM1 (**Figure 8—figure supplement 1**). The binding site of Chl *b*617 is located in a motif of the EC loop region different from the corresponding region in plant Lhcb2 or *Cr*LhcbM1. Among the seven carotenoid molecules, six of them span the membrane and connect the lumenal layer Chls with those in the stromal layer, whereas one (Prx719) lies parallel to the plane of the lumenal layer and contributes to trimerization of *Ot*Lhcp1 and *Ot*Lhcp2 (**Figure 8—figure supplement 2**). While three carotenoids (Dlt715, Dlt716, and Nex717) of *Ot*Lhcp1/2 have similarly placed counterparts in *Zm*Lhcb2 or *Cr*LhcbM1, the other four (Prx718, Prx719, Dlt720, and Dlt721) appear to be specific to *Ot*Lhcp1/2 and absent in *Zm*Lhcb2 or *Cr*LhcbM1 (**Figure 7—figure supplement 2e, f**). In addition, the violaxanthin molecule found at the trimerization interface of plant LHCII (**Liu et al., 2004**) is absent in *Ot*Lhcp1/2.

## Formation of Lhcp1-(Lhcp2)₂ and (Lhcp2)₃ trimers

Previously, it was suggested that Lhcp2 from *O. tauri* may form a hexameric unit according to the gel filtration analysis result (**Swingley et al., 2010**). The current work indicates that there are one Lhcp1 and eight Lhcp2 associated with *Ot*PSI-LHCI, forming three individual trimers (Trimers 1–3, **Figure 2a**). Among these subunits, Trimers 1 and 2 interact with each other closely and may form a stable hexameric unit. Despite that, Trimer 2 differs from the other two trimers in terms of subunit composition [Lhcp1(Lhcp2)₂ vs. (Lhcp2)₃], the overall structures of the three trimers are highly similar (**Figure 9a**). The NTR, helix C, helix E, the E-C loop, and the C-terminal region (CTR) of Lhcp1 and Lhcp2 are located at the interfaces between adjacent monomers (**Figure 9a and b**). Three carotenoids, namely Prx719, Dlt720, and Dlt721, intercalate into the space between adjacent monomers and interact closely with nearby cofactors and amino acid residues (**Figure 9b**). Prx719 spans a Trimer by extending from the periphery to the center on the lumenal side and forms hydrogen bonds and van der Waals interactions with nearby CTR amino acid residues (FKFY/FGFY motif, **Figure 8—figure supplement 1**, **Figure 8—figure supplement 2e–f**). Meanwhile, Dlt720 and Dlt721 are mainly located at the peripheral and middle regions of the trimerization interface, respectively (**Figure 9e**). Thereby, we propose that the interfacial carotenoids serve as molecular staples, holding the three monomers together and stabilizing the trimeric assembly of Lhcps.

Like *Ot*Lhcp, the major LHCII from plants and *C. reinhardtii* also exists in homotrimeric or heterotrimeric states (**Caffarri et al., 2004**; **Kawakami et al., 2019**; **Standfuss and Kühlbrandt, 2004**). Trimerization of LHCII is crucial for thermal stability of the complex, may modulate the pigment domains related to nonphotochemical quenching of Chl fluorescence, and allows the transfer of excitation energy between adjacent monomers through interfacial Chls (**Novoderezhkin et al., 2011**; **Wentworth et al., 2004**). While the *Ot*Lhcp2 and LHCII trimers share similar overall shapes, they differ

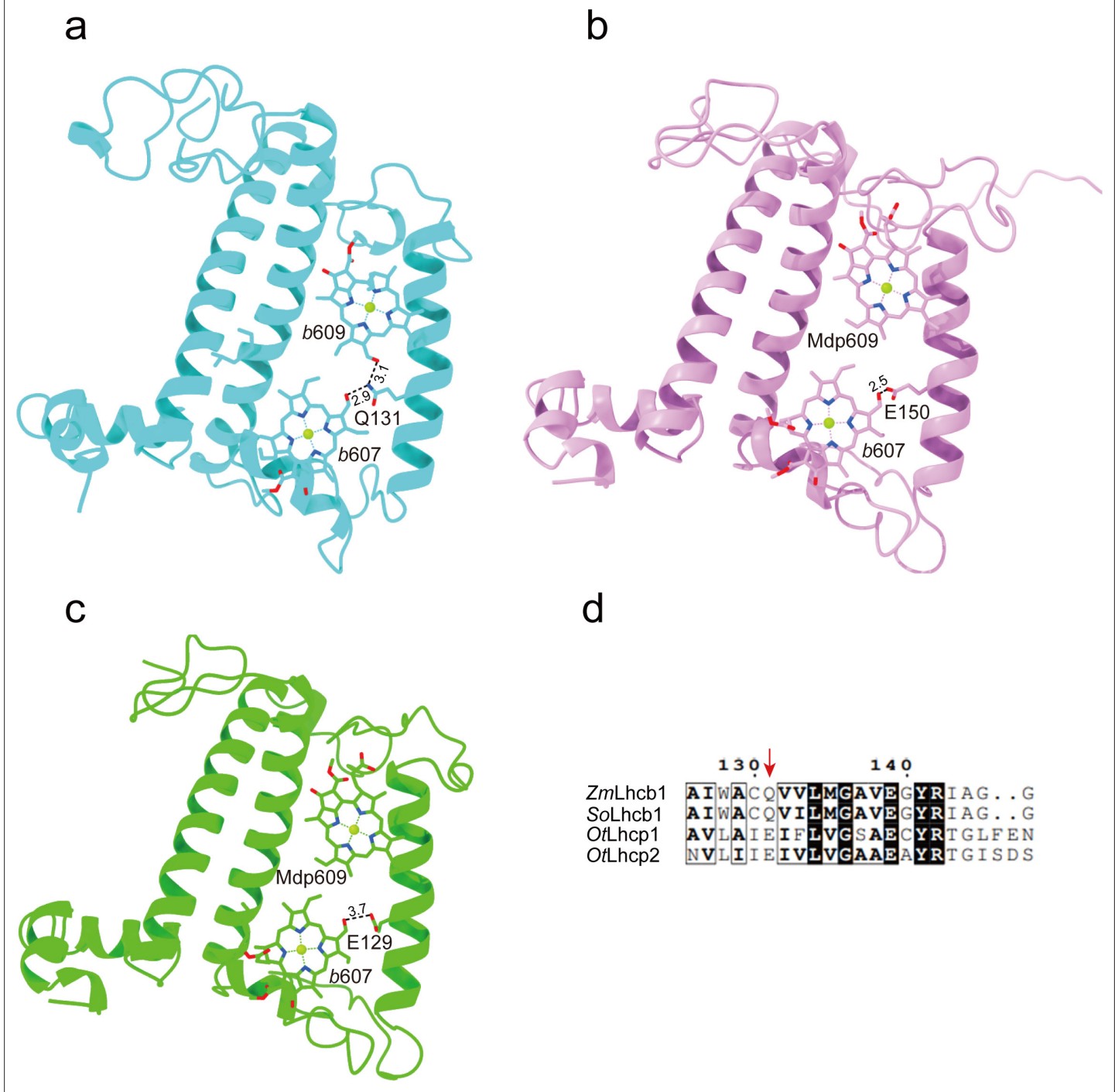

**Figure 8.** The key amino acid residues related to the selectivity of chlorophyll binding site 609 in *Ot*Lhcp1/2 and plant Lhcb1. (**a**) The side chain amine group of Gln131 in Lhcb1 from *Spinacea oleracea* (*So*Lhcb1) offers two hydrogen bond donors for selective binding of two Chl *b* molecules. (**b and c**) The carboxyl group of Glu150/Glu129 in *Ot*Lhcp1/2 provides only one hydrogen bond donor for selective binding of only one Chl *b* molecule instead of two. (**d**) Sequence alignment analysis of the plant Lhcb1 (*Z. mays* Lhcb1/*Zm*Lhcb1 and *So*Lhcb1) and *Ot*Lhcp1/2. Color codes: *cyan*, Lhcb1; *violet*, Lhcp1; *green*, Lhcp2. The dash lines indicate the putative hydrogen bonds between the C7-formyl groups of Chl *b* molecules and Glu/Gln residues. The numbers labeled nearby the dash lines are the distances (Å) between hydrogen bond donor and acceptors. The pigment molecules are shown as stick models, while the protein backbones are presented as cartoon models.

The online version of this article includes the following figure supplement(s) for figure 8:

**Figure supplement 1.** Sequence alignment of *Ot*Lhcp1 and *Ot*Lhcp2 with LhcbM1 from *C. reinhardtii* and Lhcb2 from *Z. mays*.

**Figure supplement 2.** The detailed features at the trimerization interfaces of Trimers.

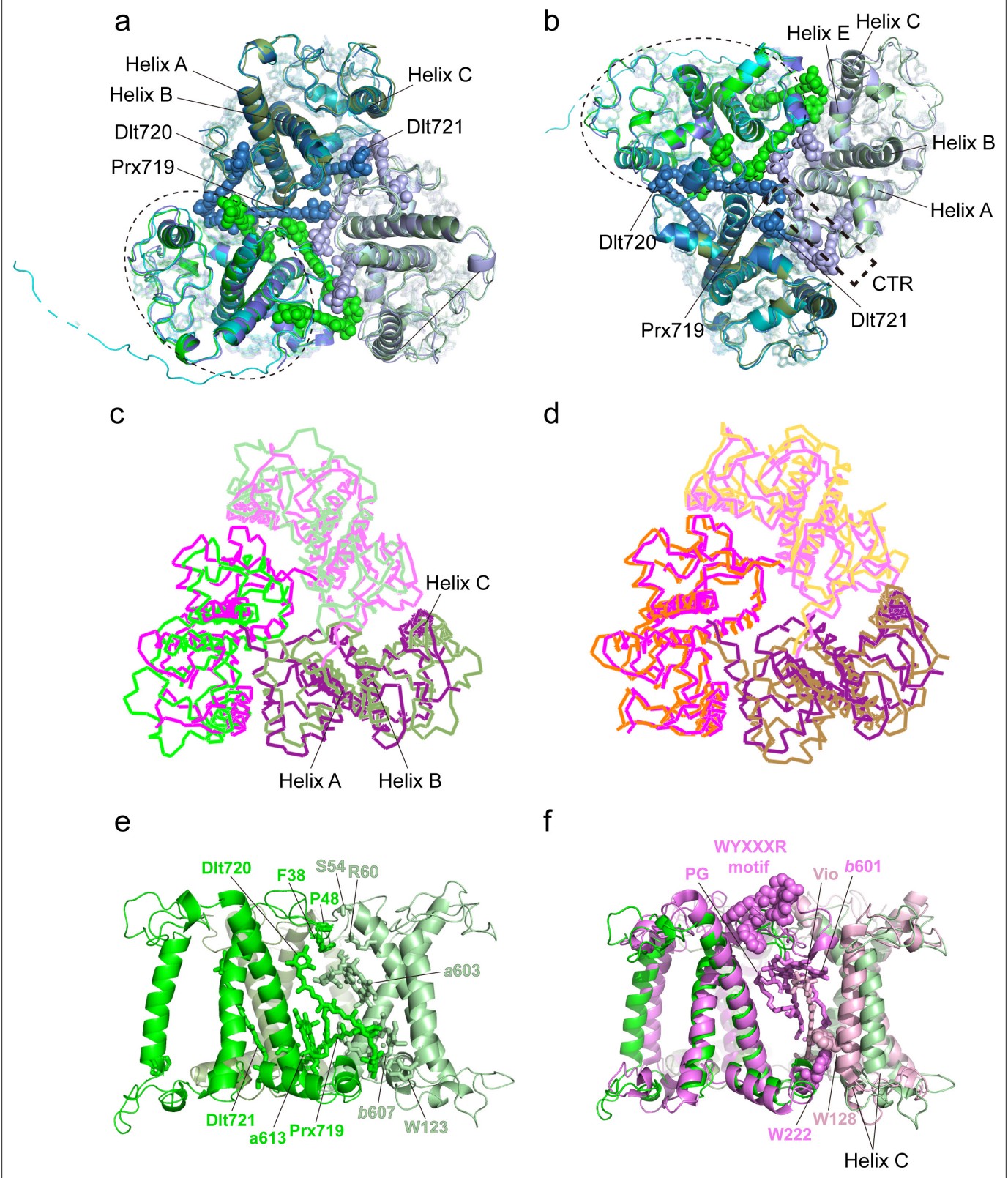

**Figure 9.** Monomer-monomer interface of the Trimers. (**a and b**) Superposition of the three Trimers associated with photosystem I (PSI) viewed from stromal (**a**) and lumenal (**b**) sides. Color codes: *green/pale green/dark green*, three different monomers of Trimer 1 [(Lhcp2)₃]; *cyan/pale cyan/dark cyan*, three different monomers of Trimer 2 [Lhcp1-(Lhcp2)₂]; *blue/light blue/dark blue*, three different monomers of Trimer 3 [(Lhcp2)₃]. The dashed elliptical ring indicates the approximate boundary of one Lhcp1/2 monomer. The dashed box outlines the interfacial location of the C-terminal regions (CTR) of

*Figure 9 continued on next page*

Figure 9 continued

Lhcp1/2. The three carotenoid molecules located at the monomer-monomer interfaces are highlighted as sphere models, while the remaining cofactors are shown as stick models. (c) The (Lhcp2)₃ trimer superposed on spinach LHCII trimer are shown as thin ribbon models. The three monomers of spinach LHCII trimer (PDB code: 1RWT) are colored in *violet*, *light pink*, and *deep purple*, respectively, whereas Lhcp2 monomers are in *green*. (d) Superposition of spinach LHCII trimer with *C. reinhardtii* LHCII trimer. The three monomers of *C. reinhardtii* LHCII (PDB code: 7D0J) are shown as thin ribbon models and colored in *orange*, *light orange*, and *sand*. (e) Side view of the monomer-monomer interface of the (Lhcp2)₃ trimer. The amino acid residues and cofactors in contact with the adjacent monomer are highlighted as sticks models. (f) The monomer-monomer interface of (Lhcp2)₃ trimer from *O. tauri* compared to the one in spinach LHCII trimer. The cofactors and amino acid residues involved in trimerization of spinach LHCII trimer, but absent in *Ot*(Lhcp2)₃ trimer, are shown as sphere models.

from each other in terms of mutual positions and distances between adjacent monomers (*Figure 9c and d*), and the interfacial cofactors and amino acid residues (*Figure 9e and f*). Besides, the phosphatidylglycerol (PG) molecule, Chl *b*601, Vio, the WYXXXR motif (also known as trimerization motif) at the NTR (*Koziol et al., 2007*) and two bulky amino acid residues (Trp128 and Trp222) found at the trimerization interface of plant LHCII trimer are all absent in the *Ot*Lhcp trimers (*Figure 8—figure supplement 1*).

## Potential transfer pathways for excitation energy between Trimers and PSI

Among the three Trimers associated with PSI, Trimer 1, formed by monomers S, T and U, may have a crucial role in mediating the excitation energy transfer between Trimer 2, formed by monomers P, Q, and R, and Trimer 3, formed by monomers V, W, and X. As shown in *Figure 10*, the Chl molecules in Trimer 1 are well connected in terms of excitation transfer with those from Trimer 2 and Trimer 3 through multiple pairs of interfacial Chls in the stromal and lumenal layers. Remarkably, the distances between adjacent Chls at the trimer-trimer interface are comparable to or even smaller than those of inter-monomer Chl pairs within the individual trimers (*Figure 10a–d*). In the stromal layer, the Chl $a611$-$a612$ dimer from monomer $S_{Trimer1}$ and the Chl $a611$-$a612$ dimer from monomer $Q_{trimer2}$ are located in close proximity to each other at 12.4 or 12.6 Å distances (Mg-Mg, *Figure 10a*). Meanwhile, Chl $a611$ from monomer $T_{Trimer1}$ and Chl $a611$ from monomer $V_{Trimer3}$ are within 18.0 Å distance. In the lumenal layer of Trimers, Chl $a614$ and $b617$ from monomer $S_{Trimer1}$ are close to Chl $b617$ and $a614$ from monomer $Q_{Trimer2}$ (at distances of 18.6 and 19.0 Å), respectively (*Figure 10b*). Meanwhile, Chl $a614$ from monomer $T_{Trimer1}$ and Chl $b605$ from monomer $U_{Trimer1}$ are in close proximity to Chl $a614$ and $b617$ from monomer $V_{Trimer3}$ at 14.4 and 9.7 Å distances, respectively. These closely positioned Chl pairs at the trimer-trimer interface may allow the excitation energy to be shared efficiently among them.

The Chls in Trimer 2 and Trimer 3 form multiple interfacial pairs with the Chls from Lhca6/PsaH and PsaK, respectively, which are likely to help provide the excitation energy for PSI (*Figure 10a and b*). In contrast, Trimer 1 appears to be poorly connected with PSI as it has Chl *b* molecules (Chls $b608$, $b605$, and $b617$ from monomer T) located at the interface with PsaO (*Figure 10a and b*) and Chl *b* molecules do not function as efficient bridges for EET between adjacent complexes (*Croce and van Amerongen, 2020*). In the stromal layer (*Figure 10a*), Chl $b608_{T/Trimer1}$ faces Chl $a2003_{PsaO}$, but they are separated at a relatively large Mg-Mg distance (23.9 Å). As the distance and orientation factor as well as the involved Chl species determine the efficiency of the inter-complex energy transfer (*Croce and van Amerongen, 2020*), three Chl pairs, namely Chl $a612_{W/Trimer3}$-Chl $a201_{PsaK}$ (0.29 ps⁻¹), Chl $a612_{R/Trimer2}$ -Chl $a610_{Lhca6}$ (0.15 ps⁻¹), and Chl $a611_{R/Trimer2}$ -Chl $a302_{PsaH}$ (0.17 ps⁻¹), are predicted to mediate the most efficient energy transfer from Trimers to PSI (*Table 5*), whereas no Chl *a* in Trimer 1 was involved. Although the distances for Chl $a610_{W/Trimer3}$-Chl $a201_{PsaK}$ (0.00 ps⁻¹) and the Chl $a611_{R/Trimer2}$-Chl $a609_{Lhca6}$ (0.05 ps⁻¹) are also short (18.5 or 20.8 Å), their energy transfer efficiency would be negligible due to the relatively low orientation factor ($K^2$ <0.7, *Figure 10—figure supplement 1*). Thus, most of the energy collected by Trimer 1 is likely transferred to Trimer 2 or Trimer 3 and then pass through Chl $a611$-Chl $a612_{R/Trimer2}$ or Chl $a612_{W/Trimer3}$ to reach PSI core (*Figure 10—figure supplement 2*). Both Trimers 2 and 3 establish close connections with PSI subunits (PsaH and PsaK) through the Chl $a611$-$a612$ clusters from monomers $R_{Trimer2}$ and $W_{Trimer3}$, respectively. In plant LHCII, the excitation energy within a trimer is mainly populated on the Chl $a610$-$a611$-$a612$ cluster in equilibrium (*Novoderezhkin et al., 2005*). The Chl $a610$-$a611$-$a612$ clusters in monomer $R_{Trimer2}$ and monomer $W_{Trimer3}$ may have a similar function in collecting energy from the adjacent Lhcp monomers and then transfer the energy

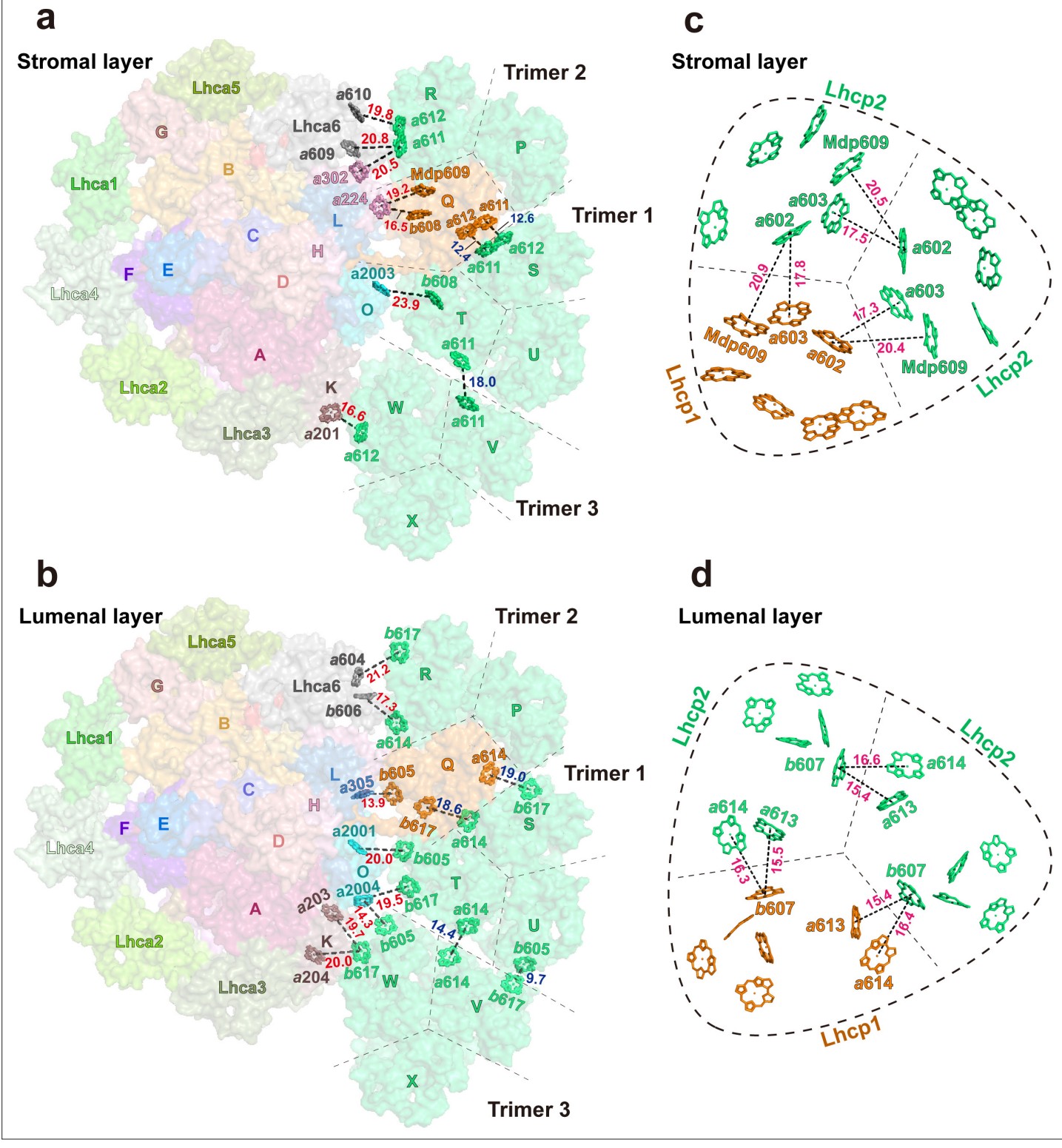

**Figure 10.** Potential energy transfer pathways between Trimers and photosystem I (PSI). (**a** and **b**) The arrangement of interfacial chlorophyll molecules within the stromal (**a**) and lumenal (**b**) layers. The view is from stromal side and approximately along membrane normal. The Chl molecules at the interfaces between adjacent Trimers and between PSI and Trimers are highlighted as stick models, while the surface presentation of the supercomplex is shown in the background. The labels of P, Q, R, S, T, U, V, W, and X indicate the chain names of Lhcp1 (Q)/Lhcp2 (P, R-X) monomers, whereas those of A-O represent the chain names of PsaA-PsaO in PSI. (**c** and **d**) The arrangement of Chl molecules in the Trimer 2 within the stromal (**c**) and lumenal (**d**) layers. The Chl molecules at the interfaces between adjacent monomers of Trimer 2 are highlighted as stick models. Color codes: *orange*, Lhcp1;

*Figure 10 continued on next page*

*Figure 10 continued*

*mint green*, Lhcp2. The *red, blue,* and *magenta* labels indicate the close relationships between two adjacent Chl molecules at the interface between Trimers 1-3 and PSI/Lhca6 subunits, between two adjacent Trimers and between two adjacent Lhcp monomers within the trimer, respectively. The Mg-Mg distances (Å) are labeled nearby the dotted lines. The dark dashed lines define the estimated interfaces between the adjacent Lhcp monomers, while the dashed round triangles outline the approximate boundary of the trimer. Chains P, Q, and R, Trimer 2; Chains S, T, and U, Trimer 1; Chains V, W, and X, Trimer 3. The phytol tails of Chl molecules are omitted for clarity.

The online version of this article includes the following figure supplement(s) for figure 10:

**Figure supplement 1.** An example of a close Chl pair that has a low orientation factor and a low FRET rate.

**Figure supplement 2.** Structure-based analysis of FRET networks within the PSI-LHC-Lhcp supercomplex.

to the nearby Chls on PsaH and PsaK. Meanwhile, Lhca6 may also serve as an intermediate complex connecting the energy transfer process from $R_{Trimer\,2}$ to PSI.

## Physiological role of the *Ot*PSI-LHCI-Lhcp supercomplex

The structure of the *Ot*PSI-LHCI-Lhcp supercomplex, which was examined in this study, reveals the association of Lhcp trimers with the PSI core and the interfacial pigment pairs, supporting their function as peripheral antennae. In higher plants and the core chlorophyte *C. reinhardtii*, their LHCII trimer(s) bound to the same side of the PSI core act as conditional antennae under state 2 conditions (*Huang et al., 2021*; *Pan et al., 2018*; *Pan et al., 2021*). However, in the present study, the *O. tauri* cells were grown under white LL conditions (50 μmol photon m$^{-2}$ s$^{-1}$) without inducing state 2 (*Minagawa,*

**Table 5.** Calculated FRET rates between Lhcp and photosystem I (PSI) core pigment pairs.
In each row of the pigment pair column, on either side of the arrow, the first letter, the second value, and the third subscript represent the Chl species, the Chl number, and the polypeptide chain, respectively.

| Pigment pair | FRET rate $k_{FRET}$ (ps$^{-1}$) | Distance R (Å) | Dipole orientation factor $K^2$ | Chl *a* – Chl *a* pair |
|---|---|---|---|---|
| b605$_Q$ → a305$_L$ | 0.27 | 13.53 | 1.00 | No |
| b605$_W$ → a2004$_O$ | 0.29 | 13.98 | 1.28 | No |
| b608$_Q$ → a224$_H$ | 0.19 | 16.56 | 2.38 | No |
| a612$_W$ → a201$_K$ | 0.29 | 16.24 | 0.95 | Yes |
| a614$_R$ → b606$_{Lhca6}$ | 0.02 | 17.39 | 0.35 | No |
| b605$_T$ → a2003$_O$ | 0.05 | 16.84 | 0.66 | No |
| a610$_W$ → a201$_K$ | 0.00 | 18.46 | 0.01 | Yes |
| Mdp609$_Q$ → a224$_H$ | 0.01 | 19.08 | 1.36 | No |
| b606$_Q$ → a224$_H$ | 0.03 | 19.28 | 0.80 | No |
| b617$_T$ → a2004$_O$ | 0.05 | 19.40 | 1.63 | No |
| a612$_R$ → b606$_{Lhca6}$ | 0.00 | 18.87 | 0.70 | No |
| b617$_W$ → a203$_K$ | 0.06 | 19.75 | 2.02 | No |
| b617$_W$ → a204$_K$ | 0.05 | 20.00 | 1.83 | No |
| a612$_R$ → a610$_{Lhca6}$ | 0.15 | 19.37 | 1.42 | Yes |
| b605$_T$ → a2001$_O$ | 0.02 | 19.67 | 0.71 | No |
| a611$_R$ → a302$_H$ | 0.17 | 20.08 | 1.99 | Yes |
| b605$_Q$ → a2005$_O$ | 0.03 | 21.20 | 1.51 | No |
| b617$_R$ → a604$_{Lhca6}$ | 0.02 | 20.83 | 0.87 | No |
| a611$_R$ → a609$_{Lhca6}$ | 0.05 | 20.79 | 0.70 | Yes |
| b617$_W$ → a206$_K$ | 0.02 | 20.86 | 0.79 | No |
| b605$_Q$ → a224$_H$ | 0.01 | 21.37 | 0.79 | No |
| b617$_W$ → a2004$_O$ | 0.00 | 21.54 | 0.01 | No |

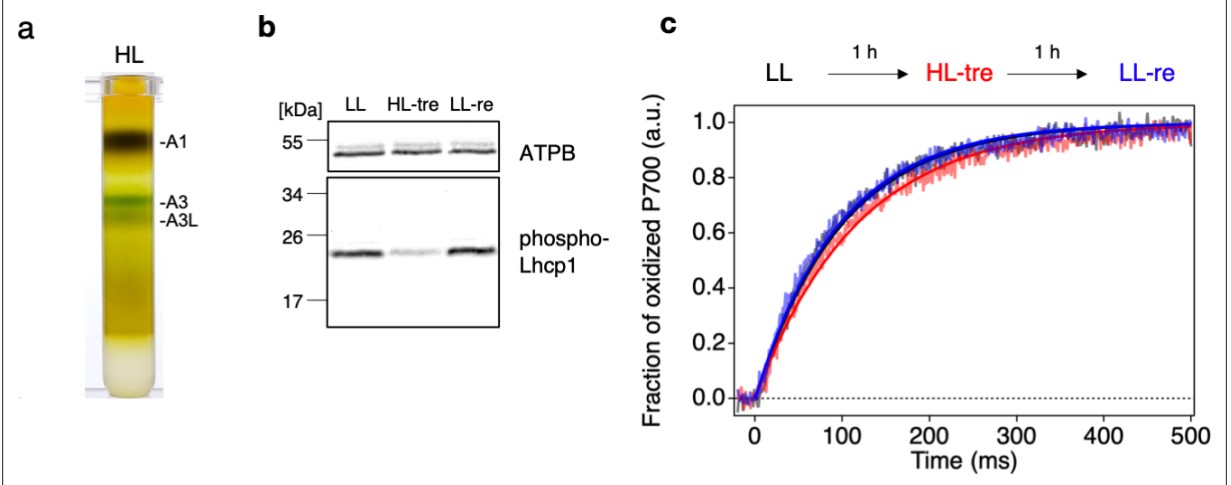

**Figure 11.** Long- and short-term acclimation of photosystem I (PSI) supercomplex in *O. tauri*. (**a**) Sucrose density gradient ultracentrifugation of the solubilized thylakoids (0.25 mg Chl) from *O. tauri* cells grown under low light (LL) and high-light (HL) conditions. (**b**) Immunoblot analysis of phospho-Lhcp1. Thylakoids (2 µg Chl) were isolated from LL-, HL-treated, and LL-recovered cultures (LL [*black*], HL-tre [*red*], and LL-re [*blue*], respectively). ATPB protein levels are shown as the loading control. Data are representative of two independent experiments. See another set of data in *Figure 11—figure supplement 1*, (**c**) Light-induced P700 oxidation kinetics measured under 22 µmol photon m$^{-2}$ s$^{-1}$ in the thylakoids sampled from the same cultures in (**b**). The fraction of P700 oxidation was derived from Δ(A$_{820}$-A$_{870}$). *Solid* lines and *shaded* lines represent fitting curves by mono-exponential functions and averaged lines of measurements of eight technical replicates, respectively. Data are representative of two independent experiments. See another set of data in *Figure 11—figure supplement 2*.

The online version of this article includes the following source data and figure supplement(s) for figure 11:

**Source data 1.** Quantitative data for *Figure 11b*.

**Source data 2.** Raw data for *Figure 11c*.

**Figure supplement 1.** Immunoblot analysis of phospho-Lhcp1.

**Figure supplement 1—source data 1.** Raw data for *Figure 11—figure supplement 1*.

**Figure supplement 2.** Light-induced P700 oxidation kinetics measured under 22 µmol photon m$^{-2}$ s$^{-1}$.

**Figure supplement 2—source data 1.** Quantitative data for *Figure 11—figure supplement 2*.

*2011*). Under LL conditions, we observed a higher amount of the A3L band compared to the A3 band (*Figure 1a*), while this difference was not observed under HL conditions (500 µmol photon m$^{-2}$ s$^{-1}$, *Figure 11a*). These findings suggest that the PSI-LHCI-Lhcp and PSI-LHCI supercomplexes may either represent the PSI-LHCI-Lhcp supercomplex before and after photodamage caused by HL treatment, or the LL- and HL-acclimated forms of the PSI supercomplex in this primordial green alga. If the latter is the case, our results may be comparable to those observed in the moss *Physcomitrella patens*, where the PSI supercomplex that is predominantly formed under HL conditions shares architectural similarities with that found in higher plants (*Yan et al., 2021*; *Gorski et al., 2022*). In contrast, the other PSI supercomplex, formed under LL or mixtrophic conditions, associates several additional LHC proteins, including an LHCII trimer, to increase the antenna size (*Iwai et al., 2018*; *Pinnola et al., 2018*). This structural arrangement is postulated as a 'constitutive state 2' (*Pinnola et al., 2018*), as the binding site of the LHCII trimer to PSI appears to be constitutively occupied, which is partly supported by the low capability of state transitions in *P. patens* (*Busch et al., 2013*). While formation of the latter large-type PSI supercomplex requires expression of a unique LHC polypeptide Lhcb9 under LL conditions, which contains red-shifted absorption forms (*Pinnola et al., 2018*), there is no counterpart of Lhcb9 found in *O. tauri*.

Alternatively, our observation of two different forms of PSI supercomplexes under LL and HL conditions may be the results of a short-term acclimation response, such as state transitions. This hypothesis is supported by a proposed mechanism for the docking and undocking of Lhcp trimers that resembles that of state transitions (*Rochaix, 2014*). Specifically, in *O. tauri*, the phosphorylated Lhcp1-(Lhcp2)$_2$ (or Trimer 2) docks to the pocket formed by PsaH/L, which shows similarities to those observed in *Z. mays* (*Pan et al., 2018*) and *C. reinhardtii* (*Pan et al., 2021*), including the crucial RRpT motif

(*Figure 6d*). During state transitions, LHCII trimers become phosphorylated when the plastoquinone pool is reduced, and a portion of the phospho-LHCII trimer(s) is associated with the pocket formed by PsaH/L on the PSI core using the RRpT motif (*Pan et al., 2021*). The architecture supporting this mechanism appears to be conserved across both branches of green plants: streptophytes including higher plants (*Pan et al., 2018*) and chlorophytes including *C. reinhardtii* (*Pan et al., 2021*), suggesting that it emerged early in the evolution of green plants. *O. tauri*, whose ancestor diversified near the branch point of chlorophytes and streptophytes, may retain some of the original features of state transitions (*Lewis and McCourt, 2004*).

To investigate whether the observed differences in the PSI supercomplex formation in *O. tauri* are due to conditional binding of Lhcp trimers, we further examined the response of the cells in the short term as shown in *Figure 11b–c*. *O. tauri* cells grown under LL conditions (LL-culture) were subjected to HL-treatment for 1 hr and then returned to LL conditions for another 1 hr of incubation (LL-recovery). The phosphorylation of Lhcp1 was detected immunologically, and the formation of PSI-LHCI-Lhcp and PSI-LHCI supercomplexes was functionally monitored by measuring the PSI antenna size, assuming that the former has a larger antenna size as seen in *Figure 1c*. This is because any biochemical manipulations may affect the supercomplex stability. The results indicate that the Lhcp1 phosphorylation was greatly reduced upon HL treatment and recovered upon returning to LL, demonstrating that Lhcp phosphorylation and dephosphorylation are reversible and regulated in the short term depending upon light intensity (*Figure 11b*, *Figure 11—figure supplement 1*). Moreover, the PSI antenna size of the LL-culture decreased upon HL treatment, although the reverse reaction was unclear, suggesting that some unknown factor(s) may be involved in the restoration reaction (*Figure 11—figure supplement 2*). These results suggest that the phosphorylation and dephosphorylation of Lhcp depend on short-term changes in light intensity, which may affect the binding of Lhcp trimers to PSI. However, whether the phospho-Lhcp trimers are constitutive antennas associated with PSI under LL conditions as in moss or conditional antennas during state transitions as in land plants or *Chlamydomonas* is currently unclear. Further investigation is needed to fully elucidate the mechanism behind the PSI antenna size regulation in *O. tauri*, which is expected to answer several emerging questions, such as whether the PSII antenna size varies complementary to that of the PSI, which kinase and phosphatase in *O. tauri* are responsible for phosphorylation and dephosphorylation of Lhcp1, and how their activities are regulated in response to light intensity change. Furthermore, if *O. tauri* in fact undergoes state transitions, it is essential to investigate the functional significance of state transitions as their PSI do not contain red chlorophylls. Determining the mode of PSI antenna size regulation in this primordial alga can thus provide valuable insights into the adaptation of photosynthesis in green plants.

In conclusion, the cryo-EM structure of the PSI supercomplex in *O. tauri* has revealed a unique hybrid of both plant-type and green algal-type PSI supercomplexes with three Lhcp trimers located at the 'state 2' position. Based on our comprehensive analysis encompassing structural, biochemical, and spectroscopic techniques, we propose that the early branching green alga *O. tauri* may have evolved a mechanism that enables it to regulate its light-harvesting capacity of PSI in response to changes in light intensity.

# Materials and methods

## Strain and growth conditions

*O. tauri* strain OTH95 (*Derelle et al., 2006*) was obtained from Roscoff Culture Collection (RCC745). Cells were grown in artificial seawater made with Sigma sea salt (Sigma cat#59883) supplemented with Daigo's IMK medium (Fujifilm-Wako, Japan) at 25°C under white fluorescence bulb (Panasonic FL40SS W/37R) at the indicated light intensities with constant air-bubbling and gentle stirring. For the short-term photoacclimation experiments, cells grown at 50 µmol photon $m^{-2}$ $s^{-1}$ (LL-culture) were shifted to 500 µmol photon $m^{-2}$ $s^{-1}$ for 1 hr (HL-treated culture), then returned to 50 µmol photon $m^{-2}$ $s^{-1}$ and incubated for 1 hr (LL-recovered culture).

## Preparations of pigment-protein complexes

Thylakoid membranes were prepared as described previously (*Swingley et al., 2010*), with a modified cell lysis protocol by using two rounds of nebulization by BioNeb (Glas-Col, Terre Haute, IN, USA) at 8 kg $cm^{-2}$ force (of $N_2$). The obtained membranes were solubilized with dodecyl-α-D-maltoside

(Anatrace, Maumee, OH, USA), which was subsequently replaced by amphipol A8-35 (Anatrace), and the pigment-protein complexes were fractionated by the SDG ultracentrifugation as described previously (*Watanabe et al., 2019*).

## SDS-PAGE and immunoblotting

SDS-PAGE and immunoblotting were performed as described previously (*Iwai et al., 2008*). Anti-phospho-Lhcb2 (AS13 2705) and anti-ATPB (AS05 085) antibodies were obtained from Agrisera (Vännäs, Sweden).

## Mass spectrometry

The sample was applied for trypsin digestion (*Shevchenko et al., 2006*) and analyzed by a UPLC system (EASY-nLC 1000, Thermo Fisher Scientific) coupled to an Orbitrap Elite mass spectrometer (Thermo Fisher Scientific). The analyzed data were submitted for protein database search using Proteome Discoverer software (Thermo Fisher Scientific) and Mascot version 2.5.1 (Matrix Science). The protein database was generated from the polypeptide sequences of *O. tauri* OTH95 deposited in NCBI (*Derelle et al., 2006*).

## Pigment analysis

Pigments were analyzed on a UPLC system as described previously (*Tokutsu and Minagawa, 2013*) with the following modifications. Pigments were extracted from the samples with 80% aceton. Separation was carried out on a Cadenza CD-C18 UP, 2×150 mm, 3 μm column (Imtakt, Kyoto, Japan). Gradient elution was established with three-solvent system, acetonitrile/isopropanol/water. The gradient was shifted from 65:15:20 (vol/vol) to 80:15:5 (vol/vol) at 6.5 min and to 50:50:0 (vol/vol) at 10 min before returning to 65:15:20 (vol/vol) at 15 min. The column temperature was 45°C. The system was calibrated with commercial standards as long as they were available (DHI, Hoersholm, Denmark). Concentrations of Dlt and Mdp were estimated based on the response factor of lutein and Chl $c_2$, respectively, and uriolide and micromonal were estimated based on the response factor of Prx as previously described (*Latasa et al., 2004*). We used extinction coefficients from *Roy et al., 2011*.

## UV-Vis absorption spectroscopy

Absorption spectra were obtained at 0.5 nm intervals in the range 400–800 nm by using a UV-Vis spectrometer V-650 (JASCO Corp, Tokyo, Japan) with an integrating sphere ISV-722 (JASCO) at room temperature. Prior to the measurements, concentration of the samples was adjusted to 3 μg Chl mL$^{-1}$.

## Fluorescence spectroscopy

Time-resolved fluorescence spectra were measured at 77 K as described previously (*Yokono et al., 2015a*), by using a photoluminescence spectrometer FLS1000-ss-stm (Edinburgh Instruments, Livingston, UK). Prior to the measurements, concentration of the samples was adjusted to 4 μg Chl mL$^{-1}$. Excitation wavelength was 405 nm. The instrumental function was measured with each sample at 405 nm with 2 nm bandwidth. Since the typical FWHM of the instrumental function was 50 ps, the theoretical temporal resolution was at most 5 ps for signals immediately after deconvolution analysis (*O'Connor and Phillips, 1984*). The excitation laser intensity was less than 8 μW with the repetition rate at 2 MHz, which did not interfere with measurements up to 100 ns. Time intervals is 2.44 ps channel$^{-1}$ for up to 10 ns and 97.7 ps channel$^{-1}$ for up to 100 ns. The optical slit was set to 10 nm and the optical filter (LOPF-25C-593) was used to cut the excitation pulse from the fluorescence signal. The signal-to-noise ratio was over 30,000:1. The fluorescence decay-associated spectra and the steady-state fluorescence spectra were constructed as described previously (*Yokono et al., 2015b*; *Yokono et al., 2019*). Briefly, the fluorescence decays were fitted using convolution and simulation method using Mathematica 12 (Wolfram Research, Champaign, IL, USA). Instrumental function and free exponential components were used to simulate fitting curves, and the determined exponential components were used to construct deconvoluted decay curves. The deconvoluted decay curves were imported to Igor 8 (WaveMetrics, Lake Oswego, OR, USA) to perform global analyses. Following a global analysis of the fluorescence kinetics, FDAS were constructed. For the PSI-LHCI and the PSI-LHCI-Lhcp supercomplexes, steady-state fluorescence spectra were reconstructed from first and second lifetime components of FDAS by integration on time axis.

## PSI antenna size measurements

Thylakoids were prepared as follows. Twenty mL of cell culture was initially shock frozen with liquid $N_2$ under the culturing light either at 50 µmol photon $m^{-2}$ $s^{-1}$ (LL-culture and LL-recovery-culture) or 500 µmol photon $m^{-2}$ $s^{-1}$ (HL-culture). The frozen cells were disrupted by adding 30 mL of disrupting buffer (25 mM HEPES [pH 7.5]), 10 mM sodium fluoride, and proteinase inhibitor PefablocSC at 0.25 mg $mL^{-1}$ (Roche, Basel, Switzerland) before centrifugation at 22,000 × $g$ for 2 min at 4°C. The pellets were resuspended in measuring buffer (25 mM HEPES [pH 7.5], 330 mM sucrose, 1.5 mM NaCl, 5 mM $MgCl_2$, 10 mM sodium fluoride) at 60 µg Chl $mL^{-1}$. A3 and A3L fractions were resuspended in measuring buffer (25 mM HEPES, pH 7.5) at 20 µg Chl $mL^{-1}$. These samples were incubated in the presence of 100 µM 3-(3,4-dichlorophenyl)-1,1-dimethylurea, 5 mM sodium ascorbate and 400 µM methyl-viologen for 5 min to create a donor-limited situation (*Melis, 1982*) and P700 oxidation kinetics were analyzed by monitoring the difference between two transmission pulse signals at 820 and 870 nm using a Dual/KLAS-NIR spectrophotometer (Heinz Walz GmbH, Effeltrich, Germany) as previously reported (*Klughammer and Schreiber, 2016*). Dark-light-induced kinetics were obtained with red actinic light (16, 28, and 48 µmol photon $m^{-2}$ $s^{-1}$, 635 nm). Traces were normalized between the minimum (before actinic light illumination) and the maximum values and fitted with a mono-exponential function to determine the P700 oxidation rate ($\tau^{-1}$).

## Negative staining single-particle analysis

Negative staining single-particle analysis was performed as described previously (*Watanabe et al., 2019*) with following modifications. Isolated protein samples were diluted to 2 µg Chl $mL^{-1}$. Electron micrographs were obtained at ×100,000 magnification and recorded at a pixel size of 5.0 Å. In total, 50 micrographs for the A3, 200 micrographs for A3L were collected. Two-dimensional (2D) classification was performed into 50 classes. Small PSI (PSI-LHCI) and large PSI (PSI-LHCI-Lhcp) supercomplexes were assigned based on the shape of the averaged particles.

## FRET calculation

FRET rate constants were computationally calculated based on the simplified FRET principle with an approximation that all Chl $a$ and $b$ had the identical excited-state energy levels so the spectral overlap integral for each combination was constant as previously described (*Mazor et al., 2017*; *Sheng et al., 2019*).

## Cryo-EM data collection, processing, classification, and reconstruction

A total of four datasets were collected on a 300 kV Titan Krios microscope (FEI) equipped with K2 camera (Gatan) by using four different grids prepared under similar conditions. Movies were captured with a defocus value at the range of –1.8 to –2.2 µm. The physical pixel size and total dose are 1.04 Å and 60 $e^-$ $Å^{-2}$, respectively. After removing images with poor quality or evident drift, a total of 19,680 (2697, 2992, 6870, and 7121 for each dataset) images were selected and used for further processing. The beam-induced motion in each movie with 32 frames were aligned and corrected using MotionCor2 (*Zheng et al., 2017*). The corrected images after alignment were used for estimation of the contrast transfer function (CTF) parameters by using CTFFIND4.1 (*Rohou and Grigorieff, 2015*). The initial image-processing steps, including manual particle picking, reference-based particle autopicking, reference-free 2D classification and 3D classification, were performed in cryo-SPARC (*Punjani et al., 2017*). The image subsets were divided into about 10 groups by reference-free 2D classification, which were further applied as templates for reference-based particle autopicking. Subsequently, the total of 5,288,217 raw particles were picked and further classified through the 2D classification procedure. As a result, 842,823 particles with good contrast were output from cryo-SPARC and imported into Relion 3.1 (*Scheres, 2012*) for further processing step. The particles were subjected to 3D classification without providing any references. The selected classes (2452, 6648, 32,751 and 38,722 particles for each of the four datasets) of PSI-LHCI-Lhcp supercomplex from 3D classification were combined and used for further processing through the 3D auto-refinement with C1 symmetry, CTF refinement, and Bayesian particles polishing. The final cryo-EM density map resolution

of *Ot*PSI-LHCI-Lhcp supercomplex was estimated by using the gold standard Fourier shell correlation (FSC) of two half maps with a cutoff at 0.143. To identify each individual subunits of the three Trimers, the local refinement procedure was carried out to further improve the map quality around each of the three Trimers. The local masks around each individual Trimers were applied during the refinement and the final resolutions of Trimers 1, 2, and 3 were 3.3, 2.9, 3.5 Å, respectively.

## Model building and refinement

To build the model of *Ot*PSI-LHCI-Lhcp supercomplex, the structure of *Zm*PSI-LHCI (PDB code: 5ZJI) was fitted into the PSI-LHCI region of the 2.94 Å cryo-EM map by using UCSF chimera and manually adjusted through rigid body fit zone in COOT to achieve improved matching of the model with density. Subsequently, the amino acid residues in the model of each chain were corrected and registered by referring to the sequences of corresponding proteins from *O. tauri*. The PsaM subunit was added manually by de novo model building with the amino acid sequence as a reference. The Lhca5 and Lhca6 model was built by referring to the corresponding ones from the *Cr*PSI-LHCI-LHCII structure (PDB code: 7D0J) and each amino acid residues in the model were checked and corrected by referring to local density feature and the *Ot*Lhca5 and *Ot*Lhca6 sequences. The density of Lhcp1 exhibits an elongated NTR similar to the corresponding part in *Cr*LhcbM1, and shows a well-defined side chain densities in the local part found in the surface binding pocket of PSI. The model of Lhcp1 was built by using *Cr*LhcbM1 (PDB code: 7D0J) as an initial model, sequence registration and correction of the model was performed by referring to the map and *Ot*Lhcp1 sequence simultaneously. For the model of Lhcp2, an initial model was adapted from an LHCII monomer, which was corrected and refined according to the cryo-EM map and *Ot*Lhcp2 sequence. For pigment molecules, two different carotenoids (Prx and Dlt) and one Chl (Mdp) found in prasinophyte were identified and assigned in each Lhcp1/2 monomer by referring to their characteristic density features. As a result, each Lhcp1/2 monomer contains four Dlt, two Prx, and one Mdp in addition to one Nex, eight Chl *a*, and five Chl *b*. Among the Chls, one (Chl *b*617) is located at the peripheral region of each Lhcp1/2. These pigment molecules show well-defined density features useful for their identification and model building. As the unambiguous identification of Chl *a* from Chl *b* requires higher resolution at around or beyond 2.8 Å, the current models of Chl *a* and Chl *b* were tentatively assigned by considering the local features and binding environments around C7 group.

After model building, real-space refinement of the *Ot*PSI-LHCI-Lhcp supercomplex structure against the 2.94 Å overall map was performed by using Phenix 1.19.2 (*Afonine et al., 2018*). The models of Trimers 1 and 3 were refined by using the 3.3 and 3.5 Å local maps, respectively. The two Lhcp2 (Chain P and Chain R) of Trimer 2 were firstly refined against the 2.9 Å local map, combined with the Lhcp1 model and then further refined against the overall map. The geometric restraints, between magnesium ions of Chl molecules or irons of $Fe_4S_4$ clusters and their coordination ligands, were applied during the refinement process. After real-space refinement, manual adjustment and correction was carried out iteratively in COOT. The geometries of the structural model were assessed using Phenix and the detailed information were summarized in *Table 3*.

## Acknowledgements

We thank Bo-Ling Zhu, Xiao-Jun Huang, Xu-Jing Li, and other staff members for their support in cryo-EM data collection at the Center for Biological Imaging (CBI), Core Facilities for Protein Science at the Institute of Biophysics, Chinese Academy of Sciences. We are grateful to Mr. Masato Kubota for his technical assistance for biochemical characterizations and insightful discussion. We also thank Mr. Hiroki Mizoguchi and Yoshihiro Kawada for their involvement in the early phase of this research, Ms. Xiao-Bo Liang for her assistance in sample shipment and handling and Dr. Mei Li for discussion. We are grateful to Mrs. Tomoko Mori and Ms. Yumiko Makino (Trans-Omics Facility, NIBB Trans-Scale Biology Center) for providing technical assistance with LC-MS/MS analysis. The project is funded by Japan Society for the Promotion of Science (JSPS) KAKENHI (21H04778 and 21H05040 to JM), the National Natural Science Foundation of China (31925024 to ZL), the Chinese Academy of Sciences (the Strategic Priority Research Program XDB37020101 and Young Scientists in Basic Research YSBR-015 to ZL) and the National Key R&D Programme of China (2017YFA0503702 to ZL). This work was also supported by Model Plant Research Facility, NIBB, and the Cooperative Study Program of National Institute for Physiological Sciences (NIPS).

## Additional information

### Competing interests
Jianyu Shan: Xin Sheng: The other authors declare that no competing interests exist.

### Funding

| Funder | Grant reference number | Author |
|---|---|---|
| Japan Society for the Promotion of Science | 21H04778 | Jun Minagawa |
| Japan Society for the Promotion of Science | 21H05040 | Jun Minagawa |
| National Natural Science Foundation of China | 31925024 | Zhenfeng Liu |
| Chinese Academy of Sciences | XDB37020101 | Zhenfeng Liu |
| Chinese Academy of Sciences | YSBR-015 | Zhenfeng Liu |
| National Key Research and Development Program of China | 2017YFA0503702 | Zhenfeng Liu |

The funders had no role in study design, data collection and interpretation, or the decision to submit the work for publication.

### Author contributions
Asako Ishii, Data curation, Validation, Investigation, Visualization, biochemical preparation and characterization of the PSI-LHCI-Lhcp supercomplex; Jianyu Shan, Data curation, Software, Investigation, Visualization, Writing – original draft, building, refinement, and analysis of the structural models; Xin Sheng, Data curation, Investigation, Software, Visualization, collection and refinement of cryo-EM data; Eunchul Kim, Data curation, Software, Formal analysis, Validation, Investigation, Visualization, Methodology, Writing – original draft, absorption spectroscopy including P700 oxidation kinetics, single particle analysis of the negative staining EM images, and computational analysis on energy transfer; Akimasa Watanabe, Conceptualization, Investigation, biochemical preparation; Makio Yokono, Investigation, Methodology, Writing – original draft, thylakoid preparation for short-term photoacclimation experiments and fluorescence spectroscopy including FDAS analysis; Chiyo Noda, Investigation, pigment analysis and negative staining-EM; Chihong Song, Investigation, prepation of cryo-EM grids; Kazuyoshi Murata, Supervision, Methodology; Zhenfeng Liu, Conceptualization, Resources, Data curation, Supervision, Funding acquisition, Visualization, Methodology, Writing – original draft, Project administration, Writing – review and editing; Jun Minagawa, Conceptualization, Resources, Data curation, Formal analysis, Supervision, Funding acquisition, Visualization, Methodology, Writing – original draft, Project administration, Writing – review and editing

### Author ORCIDs
Akimasa Watanabe (iD) http://orcid.org/0000-0001-6068-1328
Kazuyoshi Murata (iD) http://orcid.org/0000-0001-9446-3652
Zhenfeng Liu (iD) http://orcid.org/0000-0001-5502-9474
Jun Minagawa (iD) http://orcid.org/0000-0002-3028-3203

### Decision letter and Author response
Decision letter https://doi.org/10.7554/eLife.84488.sa1
Author response https://doi.org/10.7554/eLife.84488.sa2

## Additional files

### Supplementary files
• Supplementary file 1. Polypeptides in the A3 and A3L fractions as identified by MS. (a)

Polypeptides in the A3L fraction (PSI-LHCI-Lhcp supercomplex) as identified by MS analysis. The peptides were evaluated using liquid chomatography tandem mass spectrometry analysis. Detected peptides were analyzed using Mascot ver.2.7.0 (Matrix Science, London W1U 7GB, UK) and Proteome Discoverer software (Thermo Fisher Scientific). Observed: Experimental m/z value, Mr(expet): Experimental m/z transformed to a relative molecular mass, Mr(calc): Relative molecular mass calculated from the matched peptide sequence, Score: The ions score is a value of matching level between product ion peak and calculated fragment by MASCOT. (b) Polypeptides in the A3 fraction (PSI-LHCI supercomplex) as identified by MS analysis. The bands corresponding to Lhca5 and Lhca6 (Bands 2 and 1, respectively) were excised from the SDS-PAGE gel and subsequently subjected to LC-MS/MS analysis following in-gel trypsin digestion, as described in *Kubota-Kawai et al., 2019*. Detected peptides were analyzed as in (a).

• MDAR checklist

### Data availability

The atomic coordinates of the OtPSI-LHCI-Lhcp supercomplex have been deposited in the Protein Data Bank (PDB) with accession code 7YCA. The cryo-EM map of the supercomplex has been deposited in the Electron Microscopy Data Bank (EMDB) with accession code EMD-33737. The local maps and corresponding models of the three individual Lhcp trimers have also been deposited in the EMDB and PDB under accession codes of EMD-34733 and 8HG3, EMD-34735 and 8HG5, EMD-34736 and 8HG6.

The following datasets were generated:

| Author(s) | Year | Dataset title | Dataset URL | Database and Identifier |
|---|---|---|---|---|
| Shan J, Sheng X, Ishii A, Watanabe A, Song C, Murata K, Minagawa J, Liu Z | 2022 | Cryo-EM structure of the PSI-LHCI-Lhcp supercomplex from *Ostreococcus tauri* | https://www.rcsb.org/structure/unreleased/7YCA | RCSB Protein Data Bank, 7YCA |
| Shan J, Sheng X, Ishii A, Watanabe A, Song C, Murata K, Minagawa J, Liu Z | 2023 | Cryo-EM structure of the Lhcp complex from *Ostreococcus tauri* | https://www.rcsb.org/structure/8HG3 | RCSB Protein Data Bank, 8HG3 |
| Shan J, Sheng X, Ishii A, Watanabe A, Song C, Murata K, Minagawa J, Liu Z | 2023 | Cryo-EM structure of the prasinophyte-specific light-harvesting complex (Lhcp) from *Ostreococcus tauri* | https://www.rcsb.org/structure/8HG5 | RCSB Protein Data Bank, 8HG5 |
| Shan J, Sheng X, Ishii A, Watanabe A, Song C, Murata K, Minagawa J, Liu Z | 2023 | Cryo-EM structure of the prasinophyte-specific light-harvesting complex (Lhcp) from *Ostreococcus tauri* | https://www.rcsb.org/structure/8HG6 | RCSB Protein Data Bank, 8HG6 |
| Shan J, Sheng X, Ishii A, Watanabe A, Song C, Murata K, Minagawa J, Liu Z | 2022 | Cryo-EM structure of the PSI-LHCI-Lhcp supercomplex from *Ostreococcus tauri* | https://www.ebi.ac.uk/emdb/EMD-33737 | Electron Microscopy Data Bank, EMD-33737 |
| Shan J, Sheng X, Ishii A, Watanabe A, Song C, Murata K, Minagawa J, Liu Z | 2023 | Cryo-EM structure of the Lhcp complex from *Ostreococcus tauri* | https://www.ebi.ac.uk/emdb/EMD-34733 | Electron Microscopy Data Bank, EMD-34733 |
| Shan J, Sheng X, Ishii A, Watanabe A, Song C, Murata K, Minagawa J, Liu Z | 2023 | Cryo-EM structure of the prasinophyte-specific light-harvesting complex (Lhcp) from *Ostreococcus tauri* | https://www.ebi.ac.uk/emdb/EMD-34735 | Electron Microscopy Data Bank, EMD-34735 |
| Shan J, Sheng X, Ishii A, Watanabe A, Song C, Murata K, Minagawa J, Liu Z | 2023 | Cryo-EM structure of the prasinophyte-specific light-harvesting complex (Lhcp) from *Ostreococcus tauri* | https://www.ebi.ac.uk/emdb/EMD-34736 | Electron Microscopy Data Bank, EMD-34736 |

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
