## [Editor Report]

This fundamental work represents an important contribution to our understanding of the diversity of photosynthetic mechanisms across the branches of phototrophic life, with the first high-resolution structure (2.9 Å) of a photosynthetic complex from the green alga, *Ostreococcus tauri*, an ecologically important green alga utilizes a unique antenna complex, Lhcp. The evidence suggests mechanism found here is distinct from the classical antenna state transitions seen in other organisms studied thus far, expanding our knowledge of how photosynthetic systems react to changes in light conditions and leading to a new understanding of the function and evolution of light-harvesting antennas.

---

## [Decision Letter]

**Decision letter after peer review:**

Thank you for submitting your article "The photosystem I supercomplex from a primordial green alga *Ostreococcus tauri* harbors three light-harvesting complex trimers" for consideration by *eLife*. Your article has been reviewed by 3 peer reviewers, one of whom is a member of our Board of Reviewing Editors, and the evaluation has been overseen by Jürgen Kleine-Vehn as the Senior Editor. The following individual involved in the review of your submission has agreed to reveal their identity: Beverley Green (Reviewer #2).

Essential revisions:

While the structural work is strong, there are two areas that should be improved, either by adding additional analyses or by tempering the claims in the text. First, the modeling should be reassessed as described by Reviewer #1. Second, while the structural work is strong, it does not yet fully connect the new structural information to functions, leaving the definitive cause-effect relationships to be explored in future work, which might include better modeling and studies on the effects of redox regulation, physiological state of the putative state transition. This could be addressed as per the suggestions of Reviewer #3, and/or by explicitly discussing the limitations in the text.

*Reviewer #1 (Recommendations for the authors):*

cryoEM processing and Modeling: The structural model of PSI can and probably should be improved. The main indicators are the high clash score (19 is indicated in the paper while the PDB validation report shows a score of 21). Recent models of comparable size are routinely refined to a clash score of below 10. The same is true for the high number of side chain rotamers outliers which is around 3% and should be significantly lower than 0.5% (here there are also variations between the values presented in the paper and those shown in the PDB validation report). Both of these figures indicate that structural refinement is not complete. The authors used focused maps to refine the lhcp trimers, it is their decision if they want to deposit them or not, but it does make an independent evaluation of all the positions separating lhcp1 and lhcp2 impossible. The assignment made by the authors appears to be well supported in the maps, however, lysine residue side chains are frequently disordered, and having additional positions supporting the assignment is important.

Fluorescence decays measurements and red states: In their analysis of the FDAS the authors do not comment on the shape of the fastest FDAS component, which is typically affected by transfer processes, at least the A3L fastest components seem to be compatible with a transfer process. To what do they ascribe the very different shapes of the A3 and A3L fastest components? how is this related to the presence of lhcp in the supercomplex?

It's not clear to me how you can resolve a 10 ps decay component with reasonable confidence if your instrument response is 50 ps ? the authors state that the temporal resolution of their setup is 5 ps which seems a bit optimistic to me. Ideally, the authors would calculate some confidence interval around their fitted value or use appropriate language to indicate the limitation of their measurement.

In their discussion of the absence of red chlorophylls from the OtPSI the authors compare the 603/609 chlorophyll dimers in Lhca5/6 to the ones in the CrLhca2/9. They state that the red states attributed to the 603/609 chl pair in Cr may be affected by different local environments in Ot. This is certainly a possibility but given that the presence of Asn as a coordinating residue in this Chl pair is (probably mistakenly) taken as a direct indication for a strongly coupled Chl, and CrLhca2/9 were shown to contain highly coupled Chls I think it's important to fully explore this question. For example, the presence of peripheral red states may be obscured by the presence of lhcp in A3L, and it's hard to be sure about the presence of lhca5/6 in the A3 fraction. Any further evidence for the presence of lhc5/6 in the A3 fraction would go a long way to support the author's point.

Excitation energy transfer: In line 336 the authors state that some transfer rates "would be negligible due to the relatively low orientation factor.". This conclusion should take into account the variability in the position of the different trimers which may be significant given the improvements seen upon focused refinement. This can have a large effect on the orientation factors and given the relatively short distances in these instances can result in significant transfer rates.

Fast adaptation to high light: The authors clearly show that lhcp phosphorylation responds to HL treatment, the data on the possible disassociation of lhcp is not as convincing. The authors clearly treated this issue in the text.

*Reviewer #2 (Recommendations for the authors):*

Suggestions for improvement

1. The word you need is "lumenal", not "luminal"

The "lumen" (note the e) is the inside space of a thylakoid.

Please change spelling all through the manuscript and in several figures.

2. Figure 4 (b) to (g). The colored text is next to impossible to read. Please use black text and find some other way--maybe a box border to indicate which molecule is which.

3. Figure 9 (a) and (b)-I cannot see anything that looks like a "sphere model of a carotenoid. This part of the figure should be much bigger in order to show what the authors wish to show. Panels (c) and (d) have 6 things superimposed and it's just a blur.

4. Figure 10--this is where Chains P, Q, R, etc are introduced without any explanations as to why they are necessary. LumEnal, again. Please clarify in the text and in the figure legend.

5. Figure 11. Wouldn't it be more convincing to run sucrose gradients of the HL-tre and the LL-re alongside the original LL? Maybe that turned out to be too difficult.

*Reviewer #3 (Recommendations for the authors):*

Important scientific issues.

2.a. Line 388: "…although the reverse reaction, namely 389 restoration of the antenna size, was unclear, which suggests that some unknown factor(s) 390 is possibly involved in this reaction (Figure 11—figure supplement 2).."

Could it also be that the decrease in antenna size was caused by photodamage of some sort? Can this be ruled out? If not, then it would seem that concluding the phenomenon is "acclimation" is premature.

2.b. The text seems to want to infer that the phosphorylation is responsible for the observed HL changes in PSI antenna size and that this may be similar to state transitions in other species. However, as pointed out in the text, in these species, the phosphorylation is related to an increase in PSI antenna size. This, and the fact that the antenna size did not recover even though the phosphorylation state did recover, seems to support photodamage rather than a state transition type mechanism.

2.c. Line 409: Calling the observed structures "state 1" and "state 2" is likely to cause confusion given that an actual state transition is not established and that even if the mechanism is similar, the fact that the "directionality" of the effect is not clear.

2.d. The effects on PSI antenna size are quite small, and one wonders what its significance might actually be.

2.e. If the structural changes operate in a mode similar to state transitions in other species, where the antenna move between PSI and PSII, the overall light gathering capacity does not change, only the fraction delivered to PSI compared to PSII. If this is the case, then excitation balancing rather than overall light capture seems to be more important. This is seen to be important under altered light qualities or when the relative rates of plastoquinone reduction and reoxidation are differently impacted. Did the current group text for effects of inhibitors (e.g., DCMU, DBMIB), far red light, anoxia, metabolic status, etc. on the putative state transition? Without these studies, the current study cannot distinguish between these possibilities, as this and other groups have done to elucidate similar mechanisms. Without these additional results, the study cannot strongly indicate what the function of the changes in structure or PSI antenna size.

3. What is the certainty of assignments like the following on line 608: "The site is occupied by an Mdp molecule in OtLhcp1/2…" Overall, the text does not really indicate how certain these assignments are.

4. Line 326: Why would there be "poorly connected" trimers at all? Is this an artifact or is "nature being wasteful"? If the latter, what are the physiological implications of such a poorly connected antenna?

5. Line 379, The phosphorylation of Lhcp1 was immunologically detected and the deformation of PSI-LHCI-Lhcp supercomplex was monitored by analyzing the decrease of the PSI antenna size."

The connection between the changes in PSI antenna size and "deformation" is only hypothetical at this point because the structures under the same sets of conditions were not also assessed.

---

## [Author Response]

Essential revisions:While the structural work is strong, there are two areas that should be improved, either by adding additional analyses or by tempering the claims in the text. First, the modeling should be reassessed as described by Reviewer #1.

We worked on further structure refinement, resulting in a sufficient quality. Please examine the details in the following sections. These are now described in text too, and the new PDB validation report is attached.

Second, while the structural work is strong, it does not yet fully connect the new structural information to functions, leaving the definitive cause-effect relationships to be explored in future work, which might include better modeling and studies on the effects of redox regulation, physiological state of the putative state transition. This could be addressed as per the suggestions of Reviewer #3, and/or by explicitly discussing the limitations in the text.

Thank you for the insightful comments. We completely agree with this. We mostly followed the suggestions of Reviewer #3 and also explicitly discussed the limitations of this study in the text as explained as follows.

Reviewer #1 (Recommendations for the authors):cryoEM processing and Modeling: The structural model of PSI can and probably should be improved. The main indicators are the high clash score (19 is indicated in the paper while the PDB validation report shows a score of 21). Recent models of comparable size are routinely refined to a clash score of below 10. The same is true for the high number of side chain rotamers outliers which is around 3% and should be significantly lower than 0.5% (here there are also variations between the values presented in the paper and those shown in the PDB validation report). Both of these figures indicate that structural refinement is not complete. The authors used focused maps to refine the lhcp trimers, it is their decision if they want to deposit them or not, but it does make an independent evaluation of all the positions separating lhcp1 and lhcp2 impossible. The assignment made by the authors appears to be well supported in the maps, however, lysine residue side chains are frequently disordered, and having additional positions supporting the assignment is important.

We thank the reviewer for checking the statistics of our structural model. According to your suggestion, we have refined our structural model to avoid the non-covalent and non-coordination close contacts. The rotamer parameters have also been improved. The improved model has a clash score at 5.41 and contains no rotamer outliers (0%).

To provide further evidence to support the assignment of Lhcp1 and Lhcp2, we have made new figures to show two additional sites with distinct features that distinguish Lhcp1 from Lhcp2. The new figures have been included in the updated version of Figure 2—figure supplement 3a-f. We have also deposited the models for the three individual Lhcp trimers along with their maps, so that they can become available to the readers to evaluate our results independently. The accession codes have been included in the manuscript (L.650-652). Thanks a lot for your suggestion.

**Author response image 1. sa2fig1:** Statistics of the updated structural model. The statistical data were reported by the MolProbity in Phenix program.

Fluorescence decays measurements and red states: In their analysis of the FDAS the authors do not comment on the shape of the fastest FDAS component, which is typically affected by transfer processes, at least the A3L fastest components seem to be compatible with a transfer process. To what do they ascribe the very different shapes of the A3 and A3L fastest components? how is this related to the presence of lhcp in the supercomplex?

This is a very important point and was not fully discussed in the previous text. Thank you for pointing this out. The positive peak in the fastest lifetime component observed in A3 was similar to the one observed in the cyanobacterial PSI core previously (e.g., Mimuro et al., 2010), reflecting the rapid energy transfer process from Chl near P700 to P700 and subsequent charge separation. In A3L, however, the fastest lifetime component showed a different shape from that in A3, indicating that energy transfer from Chls further away from P700 became dominant and the subsequent trapping process was masked. This indeed suggests the presence of additional antenna(s) in A3L, namely Lhcps, which are not present in A3. We now discuss this important point in the text (L.108-123).

It's not clear to me how you can resolve a 10 ps decay component with reasonable confidence if your instrument response is 50 ps ? the authors state that the temporal resolution of their setup is 5 ps which seems a bit optimistic to me. Ideally, the authors would calculate some confidence interval around their fitted value or use appropriate language to indicate the limitation of their measurement.

The theoretical time resolution after deconvolution is estimated to be 5 ps, which is 1/10th of the half-width of the instrument response function (O'Connor and Phillips, 1984, N.B. This reference is now cited in the text.). Author response image 2 demonstrates that 5 ps falls within our instrument's resolution, showcasing the results of deconvolution analysis of the raw 683 nm decay curve of A3 (red) with (left) and without (right) the 5 ps component. As indicated by the arrow, the shape of the fluorescence decay curve cannot be well-convoluted in the early phase without the 5 ps component (right), but it is well-convoluted when we introduce the 5 ps component (left). Thus, we are confident that the 5 ps lifetime component was necessary in this case. However, we are not claiming that the 5 ps theoretical resolution can be used after global analysis. The theoretical limit can only be used immediately after deconvolution analysis of the original decay curves. Thus, in the A3 FDAS obtained by global analysis of the multiple (deconvoluted) curves, the lifetime value of the first lifetime component was expressed as <10 ps, not 4.884 ps. We have followed the reviewer's suggestion and corrected the wording of the temporal resolution to avoid reader's confusion. (L520-523).

**Author response image 2. sa2fig2:** Comparison of deconvolution analysis of the raw 683 nm decay curve of A3 (red) with (left) and without (right) the 5 ps component.

In their discussion of the absence of red chlorophylls from the OtPSI the authors compare the 603/609 chlorophyll dimers in Lhca5/6 to the ones in the CrLhca2/9. They state that the red states attributed to the 603/609 chl pair in Cr may be affected by different local environments in Ot. This is certainly a possibility but given that the presence of Asn as a coordinating residue in this Chl pair is (probably mistakenly) taken as a direct indication for a strongly coupled Chl, and CrLhca2/9 were shown to contain highly coupled Chls I think it's important to fully explore this question.

Thank you very much for your suggestion. As the reviewer pointed out, the axial ligand of Chl a603/*a*609 in *O. tauri* Lhca5/6 appears to be the same as the corresponding one of Chl *a*603/*a*609 in *C. reinhardtii* Lhca9/2. Therefore, we think that only the presence of Asn residues at the axial ligand site is not sufficient to cause the spectral red form of Chl *a*603/*a*609 in this case. The local environments around chlorophylls may also induce structural and spectral changes of chlorophyll molecules. Therefore, we have explored the local environments around Chl *a*603/609 and found that Tyr69(Y69) and Tyr75(Y75) in *Ot*Lhca5 and *Ot*Lhca6 form van der Waals contacts with Chl *a*609_Lhca5_ and Chl *a*609_Lhca6_ respectively (Figure 4f and g). In comparison, these residues are replaced by tryptophan residues (Trp65/W65) in *C. reinhardtii* Lhca9 and Lhca2. Previously, it was reported that tryptophan residues located nearby the chlorophyll molecule may induce deformation of the tetrapyrrole macrocycle and cause red shift in the *Qy* absorption bands of Chls (Bednarczyk, D. et al. *Angew Chem. Int. Ed. Engl.*, 55:6901-6905, 2019). Mutation of a tryptophan residue (in van der Waals contacts with a bacteriochlorophyll) to Tyr or Phe caused a blue shift of the *Qy* absorption peak in the core light-harvesting complex of *Rhodobacter sphaeroids* (Sturgis, J. N. et al. *Biochemistry*, 36:2772-2778, 1997). Therefore, the occurrence of Tyr69 or Tyr75 residues (instead of Trp residues) around Chl *a*609_Lhca5_ or Chl *a*609_Lhca6_ may explain (at least in part) the lack of red spectral forms in the *Ot*PSI–LHCI–Lhcp supercomplex. We have added the new discussion in L174-193 of the revised manuscript.

Besides the Chl *a*603-*a*609 pair, Chl *a*611-*a*612 is the other pair of strongly coupled chlorophylls in *Cr*Lhca2/9 (Suga, M. et al. *Nat. Plants*, 2019; Pan, X. et al. *Nat. Plants*, 2021; Naschberger, A. et al. *Nat Plants*, 2022;). Unfortunately, Chl *a*611 and *a*612 molecules are not observed in Lhca5 or partly observed in Lhca6 (a612 is observed, but a611 is absent) (Author response image 3 and b). Nevertheless, the axial ligands of Chl *a*612 in both Lhca5 and Lhca6 are both Asn residues (Asn167 in Lhca5 and Asn176 in Lhca6), the same as the corresponding ones in *Cr*Lhca2/9. The axial ligands of Chl *a*611 in *Cr*Lhca2/9 are the phosphate groups of phosphatidylglycerol molecules, which form a salt bridge with a lysine residue nearby (Lys159 in Lhca2 and Lys160 in Lhca9). The corresponding ones in *Ot*Lhca5/6 are also conserved (Lys166 in Lhca5 and Lys175 in Lhca6), potentially providing the binding sites for phosphatidylglycerol molecules (which might be lost during purification) (Author response image 3 and d ). Chl *a*611 and/or *a*612 of Lhca5 and Lhca6 might be lost during purification process. Nevertheless, there are no direct experimental evidence available to support the functional role of Chl *a*611-*a*612 in contributing to the spectral red form of *Cr*Lhca2/9 (as far as we know), even though the corresponding ones were previously assigned as the red-most chlorophylls in LHCII (Remelli, R. et al. *J. Biol. Chem.* 274, 33510-33521, 1999; Rogl, H. & Kühlbrandt, W. *Biochemistry* 38, 16214-16222, 1999). Therefore, it might be better to focus on the Chl *a*603-*a*609 pair in the present work, while leaving the works on exploring the function of Chl *a*611-*a*612 in *Ot*Lhca5/6 or *Cr*Lhca2/9 for future work.

**Author response image 3. sa2fig3:** Comparison of Chl *a*611-*a*612 dimer in Lhca5/6 and Lhca9/2 complexes from *O. tauri* and *C. reinhardtii respectively*. a and b. Comparison of the Chl *a*611-*a*612 dimers and their local environments in *Ot*Lhca5/*Cr*Lhca9 subunit (a) and *Ot*Lhca6/*Cr*Lhca2 subunit. The protein and cofactors are shown as sticks model. Color code: pink, *O. tauri*; light blue, *C. reinhardtii*. Note that Chl *a*611 and *a*612 are not observed in *Ot*Lhca5, while Chl *a*611 is not observed in *Ot*Lhca6 but Chl *a*612 of *Ot*Lhca6 can be located. PG, phosphatidylglycerol. c and d. Sequence alignment analysis of Lhca5/9 (c) and Lhca6/2 (d) from the *O. tauri* and *C. reinhardtii*. The red arrow indicates the conserved residues serving as the axial ligands for Chl *a*612. The light blue arrow indicates the lysine residues involved in binding the phosphate group of the PG molecule (PG622) which serves as the axial ligand of Chl *a*611.

For example, the presence of peripheral red states may be obscured by the presence of lhcp in A3L, and it's hard to be sure about the presence of lhca5/6 in the A3 fraction. Any further evidence for the presence of lhc5/6 in the A3 fraction would go a long way to support the author's point.

Thank you for your insightful comments. Swingley et al. (2010) previously showed that their A3 sample contained Lhca5/6, and the newly added mass spectrometry results in the revised manuscript also show that the OtA3 sample in this study contains Lhca5/6 (Appendix 1-table 2). Based on our results, Chl 603-609 in Ot Lhca5/6 do not appear to be strongly coupled although their ligands for Chl 603 are Asn. This is, therefore, likely due to the *O. tauri*-specific local environment as proposed above. It is theoretically possible that it is also affected by Lhcp in A3L, but given that the red state is rather enhanced in the PSI-LHCI-LHCII sample from *Chlamydomonas*, whose LHCII-1 and LHCII-2 are bound at a positions similar to Trimer-1 and Trimer-2 in *Ostreococcus*, we refrain from making a further claim.

Excitation energy transfer: In line 336 the authors state that some transfer rates "would be negligible due to the relatively low orientation factor.". This conclusion should take into account the variability in the position of the different trimers which may be significant given the improvements seen upon focused refinement. This can have a large effect on the orientation factors and given the relatively short distances in these instances can result in significant transfer rates.

The Reviewer’s point is correct. While the positions of the three different Lhcp trimers are variable relative to the PSI-LHCI region, the chlorophylls at their interface are better-defined than those in the peripheral region, as they are restrained by adjacent molecules so that their variability is lower than the peripheral ones. After improving the structure, we revisited this issue in the revised manuscript. The relative positions of the chlorophylls did not change significantly, and the final conclusion remained the same: these energy transfer pathways are not dominant (see Figure 10—figure supplement 2). The orientation factors in the revision were more precise and are listed in Table 1. The refined coordinates of Chl a609_Lhca6_ and Chl a611_R/Trimer2_ are reliable enough to calculate the orientation factor because they correspond well with the cryo-EM maps (see Figure 10—figure supplement 1). The transition dipoles, calculated based on the coordinates of NB and ND atoms, clearly indicate a perpendicular orientation, which has a low orientation factor and causes a low FRET rate.

Fast adaptation to high light: The authors clearly show that lhcp phosphorylation responds to HL treatment, the data on the possible disassociation of lhcp is not as convincing. The authors clearly treated this issue in the text.

Thank you for clarifying this point. In the revised text, we explicitly stated that the possible dissociation of Lhcp is unclear. We further discussed that this issue needs to be clarified as part of a larger problem, specifically whether the Lhcp trimers are constitutive antennas associated with PSI under LL conditions (as observed in moss) or conditional antennas during state transitions (as observed in land plants and *Chlamydomonas*).

Reviewer #2 (Recommendations for the authors):Suggestions for improvement1. The word you need is "lumenal", not "luminal"Luminal is a barbiturate drug--see Wilipedia!!The "lumen" (note the e) is the inside space of a thylakoid.Please change spelling all through the manuscript and in several figures.

According to the Merriam-Webster dictionary, “luminal” is a variant of “lumenal” and has been frequently used in the photosynthesis literatures (https://www.merriam-webster.com/medical/luminal). As the reviewer pointed out, “Luminal” (with capital L) stands for the trademark of a drug named phenobarbital. Thank you for clarifying this point! All the “luminal” words in the updated manuscript have been replaced with “lumenal” to avoid ambiguity.

2. Figure 4 (b) to (g). The colored text is next to impossible to read. Please use black text and find some other way--maybe a box border to indicate which molecule is which.

Thank you for your suggestion. We have updated Figure 4b-g by using the labels with dark texts and colored box borders. The labels in three different colored boxes correspond to the amino acid residues from three different structures from *O. tauri*, *C. reinhardtii* and *P. sativum* respectively. The rainbow surface background has been omitted to show the target molecules/residues more clearly in the white background.

3. Figure 9 (a) and (b)-I cannot see anything that looks like a "sphere model of a carotenoid. This part of the figure should be much bigger in order to show what the authors wish to show. Panels (c) and (d) have 6 things superimposed and it's just a blur.

The depth cueing applied in the previous version of Figure 9a and b introduced a strong fog blurring the carotenoid molecules at the monomer-monomer interface. To show them more clearly, we have adjusted the depth cueing setting to reduce fog intensity and shown all nine interfacial carotenoids of interest, namely Prx719, Dlt720, Dlt721 molecules in each individual monomers of the trimer. Moreover, the images of the trimer have been enlarged to show the details better. For panels c and d, we have changed the cartoon models to the thinner ribbon models and left out the cofactors (chlorophylls, carotenoids and lipids), so that they can exhibit clearer protein backbone differences between the *O. tauri* (Lhcp2)_3_ timers and *spinach* LHCII trimer (c) or between *spinach* and *C. reinhardtii* LHCII trimer (d, as a control for comparison with c). The updated Figure 9 has been uploaded along with the revised manuscript (L305-336).

4. Figure 10--this is where Chains P, Q, R, etc are introduced without any explanations as to why they are necessary. LumEnal, again. Please clarify in the text and in the figure legend.

We have included the detailed explanations about the chain IDs for P, Q, R, etc in the legend of Figure 10 (L.873-874) as well as in the main text (L.339-344), and “luminal” has been replaced with “lumenal” throughout the updated manuscript.

5. Figure 11. Wouldn't it be more convincing to run sucrose gradients of the HL-tre and the LL-re alongside the original LL? Maybe that turned out to be too difficult.

Yes, it would be convincing. Unfortunately, however, this experiment is technically very challenging at present. Although sucrose density gradient is a material-demanding experiment using a big centrifuge rotor and so on, the precise PQ redox state, which affects the phosphorylation state of Lhcp as well as the formation of PSI-LHCI-Lhcp supercomplex, would be easily scrambled during such biochemical manipulations.

Reviewer #3 (Recommendations for the authors):Important scientific issues.2.a. Line 388: "…although the reverse reaction, namely 389 restoration of the antenna size, was unclear, which suggests that some unknown factor(s) 390 is possibly involved in this reaction (Figure 11—figure supplement 2).."Could it also be that the decrease in antenna size was caused by photodamage of some sort? Can this be ruled out? If not, then it would seem that concluding the phenomenon is "acclimation" is premature.

In biology, the inhibition of photosystems, which involves photodamage, can also be interpreted as an acclimation process. This process may be "programmed" so as to protect downstream components from stress. Initially, we used the term "acclimation" with this idea in mind. However, following Reviewer-3's feedback, we have revised the manuscript and changed the terminology (L396-399, p24 ).

2.b. The text seems to want to infer that the phosphorylation is responsible for the observed HL changes in PSI antenna size and that this may be similar to state transitions in other species. However, as pointed out in the text, in these species, the phosphorylation is related to an increase in PSI antenna size. This, and the fact that the antenna size did not recover even though the phosphorylation state did recover, seems to support photodamage rather than a state transition type mechanism.

As noted by Reviewer-3, it has not been demonstrated that the binding of Lhcp trimers to PSI is reversible in the short term. It is also possible that what we are observing is the displacement of photodamaged Lhcp trimers. By revising the manuscript, we have highlighted this point as the first item and have discussed it in relation to the case of *Physcomitrella patens* (L399-411, p24-25). In this case, additional LHC polypeptides, including an LHCII trimer, are believed to form a constitutive LL antennae. Thanks to the suggestion, our manuscript has received a more in-depth discussion.

2.c. Line 409: Calling the observed structures "state 1" and "state 2" is likely to cause confusion given that an actual state transition is not established and that even if the mechanism is similar, the fact that the "directionality" of the effect is not clear.

Again, taking into account the points above, the choice of terms was not appropriate; please see the revised manuscript, where the choice of the terms has been carefully changed (Paragraph: Physiological role of the OtPSI-LHCI-Lhcp supercomplex, p.24-28.)

2.d. The effects on PSI antenna size are quite small, and one wonders what its significance might actually be.

We do not have an answer to this question. It is possible that the observed changes in HL/LL represent only a partial aspect of the overall process, and there may be other, more substantial changes under fluctuating light conditions that have yet to be identified. For instance, the Lhcp trimers associated with PSI may serve as a platform to recruit more peripheral antenna complexes which may further expand the antenna size of PSI. A mutant strain lacking the ability to conduct this phosphorylation could provide further insight, though the matter remains open for discussion. However, it is worth noting that even if a mutant strain is obtained, the resulting phenotype may not be as striking as those observed in other organisms. Ultimately, the findings of this study present an opportunity to reconsider the significance of LHCII trimer phosphorylation within a broader context.

2.e. If the structural changes operate in a mode similar to state transitions in other species, where the antenna move between PSI and PSII, the overall light gathering capacity does not change, only the fraction delivered to PSI compared to PSII. If this is the case, then excitation balancing rather than overall light capture seems to be more important. This is seen to be important under altered light qualities or when the relative rates of plastoquinone reduction and reoxidation are differently impacted. Did the current group text for effects of inhibitors (e.g., DCMU, DBMIB), far red light, anoxia, metabolic status, etc. on the putative state transition? Without these studies, the current study cannot distinguish between these possibilities, as this and other groups have done to elucidate similar mechanisms. Without these additional results, the study cannot strongly indicate what the function of the changes in structure or PSI antenna size.

Currently, we have no experimental evidence available to argue about the points raised by the reviewer, because it is technically challenging to carry out the measurements on the two photosystems from *O. tauri* with very small excitation bias. Even if it were found that the LHC trimers move back and forth between the two photosystems in a short period of time without photodamage, the big question would now be why it has to do so. We ponder this question every day as well, but there is no answer. What we do know is that the lack of an obvious red Chl in the PSI-LHCI complex from this organism does lead to a very small excitation bias between the two photosystems in the IR excitation, whereas higher plants and core chlorophytes exhibit much larger excitation bias between the two photosystems. As described above, this issue needs to be clarified as part of a larger problem, specifically whether the Lhcp trimers are constitutive antennas associated with PSI under LL conditions (as observed in moss) or conditional antennas during state transitions (as observed in land plants and *Chlamydomonas*).

3. What is the certainty of assignments like the following on line 608: "The site is occupied by an Mdp molecule in OtLhcp1/2…" Overall, the text does not really indicate how certain these assignments are.

Mg-2,4-divinyl-phaeoporphyrin a5 monomethyl ester (Mg-DVP/Mdp) is a Chl *c*–like pigment with a free carboxyl group on the side chain of porphyrin ring IV, whereas Chl *a* or Chl *b* has a hydrophobic phytol chain covalently linked to the carboxyl group through an ester bond (Author response image 4). As shown in Author response image 4, the Mdp609 does not have an elongated tube-like density attached to the side chain of ring IV, suggesting that it is not esterized by a phytol chain. Besides, there is a positively-charged Lys77 residue in Lhcp1 (Author response image 4) and corresponding Lys56 residue in Lhcp2 (Author response image 4) nearby the carboxyl group of Mdp609, forming a salt-bridge with the carboxyl group at a bond length of about 2.8 Å. Therefore, both lines of evidence support the assignment of the site of 609 in *Ot*Lhcp1/2 as being occupied by a Mdp molecule. We have included the density of Mdp in Figure 2—figure supplement 2 and cited it in the main text (L271-274, p.17).

**Author response image 4. sa2fig4:** Cryo-EM density of Mdp609 and its interaction with nearby Lys77/56 of *Ot*Lhcp1/2 respectively. residue. a, The chemical structures of Mdp and Chl *a*. The figure is adapted from Jones, O. T. G. *Biochem. J.* 89, 182 (1963). b and c, Cryo-EM density of Mdp609 in Lhcp1 (c) / Lhcp2 (d) and nearby Lys residues. The number labeled nearby the dash line indicate the distance (Å) between the adjacent group of Mdp609 and Lys77/56.

4. Line 326: Why would there be "poorly connected" trimers at all? Is this an artifact or is "nature being wasteful"? If the latter, what are the physiological implications of such a poorly connected antenna?

Trimer 1 appears to be poorly connected with PSI, because their interface involves mainly Chl *b* molecules (*b*608, *b*605 and *b*617) from monomer T of Trimer 1 which are connected with Chl *a* molecules (*a*2003, *a*2001 and *a*2004) from PsaO. According to a recent work on light harvesting in oxygenic photosynthesis, the excitation energy transfer (EET) between light-harvesting complexes does not proceed through Chl *b* molecules, as they do not function as efficient bridges for EET between adjacent complexes (Croce and Amerongen, *Science*, 369: eaay2058, 2020). This is mainly because Chl *b* is not a good energy acceptor but an excellent excitation energy donor for Chl *a*, and its lowest energy transition is at higher energy than that of Chl *a* (Croce and Amerongen, *Science*, 369: eaay2058, 2020). On the other hand, despite that Trimer 1 is poorly connected with PSI, it is connected closely with Trimer 2 (monomer Q) and Trimer 3 (monomer V) through multiple pairs of Chl *a* molecules at their interfaces. It is noteworthy that the distances between adjacent Chl molecules at the Trimer 1-Trimer2 or Trimer 1-Trimer 3 interface from 9.7 to 19.0 Å, close to or even smaller than those of inter-monomer Chl pairs within the individual trimers (from 15.4 to 20.9 Å) (Figure 10a-d). Therefore, the energy collected by Trimer 1 may be mostly transferred to Trimer 2 and Trimer 3, and then further to PSI. Such an indirect energy transfer route from Trimer 1 to PSI (via Trimer 2 or Trimer 3) may serve to expand the light-harvesting capacity of PSI on one hand. The indirect EET route may reduce the chance of overloading the PSI reaction center and avoid the photodamage caused by the excess energy on the other. In case that the excitation energy collected by Trimer 1 (and Trimers 2 & 3) could not be converted by the reaction center in time, the excess energy may accumulate in the Lhcp complexes and become dissipated safely as heat through the non-photochemical quenching mechanism.

5. Line 379, The phosphorylation of Lhcp1 was immunologically detected and the deformation of PSI-LHCI-Lhcp supercomplex was monitored by analyzing the decrease of the PSI antenna size."The connection between the changes in PSI antenna size and "deformation" is only hypothetical at this point because the structures under the same sets of conditions were not also assessed.

Precisely, that is correct. Thank you for pointing that out. We rephrased the wording here (L.432-436).